# Augmented tactile-perception and haptic-feedback rings as human-machine interfaces aiming for immersive interactions

Zhongda Sun [1,2,3,4], Minglu Zhu[1,2,4], Xuechuan Shan [3,5] & Chengkuo Lee [1,2,3,4,6] ✉

Advancements of virtual reality technology pave the way for developing wearable devices to enable somatosensory sensation, which can bring more comprehensive perception and feedback in the metaverse-based virtual society. Here, we propose augmented tactile-perception and haptic-feedback rings with multimodal sensing and feedback capabilities. This highly integrated ring consists of triboelectric and pyroelectric sensors for tactile and temperature perception, and vibrators and nichrome heaters for vibro- and thermo-haptic feedback. All these components integrated on the ring can be directly driven by a custom wireless platform of low power consumption for wearable/portable scenarios. With voltage integration processing, high-resolution continuous finger motion tracking is achieved via the triboelectric tactile sensor, which also contributes to superior performance in gesture/ object recognition with artificial intelligence analysis. By fusing the multimodal sensing and feedback functions, an interactive metaverse platform with cross-space perception capability is successfully achieved, giving people a face-to-face like immersive virtual social experience.

The newly proposed concept "Metaverse" refers to a network of 3D virtual worlds that utilize virtual reality (VR) technology to enhance the connections between real and cyber spaces[1]. The VR-enabled digitalized world can bring users an immersive experience of interacting with reality by simulating human sensations, showing great development prospects in the area of social, education, gaming, training, rehabilitation, etc[2]. To establish an immersive VR system, besides visual stimulus enabled by head-mounted displays[3], other wearable devices, i.e., data gloves[4], VR suits[5], etc., that can simultaneously sense human motion and simulate human sensation, are experiencing significant attention recently for full-body

perception and feedback to further bridge the physical and cyber world[6].

The sensory and motor nerves of the hand are more complex than other body parts, providing fingers with high flexibility to realize complex interactions in VR systems. Current mature sensing techniques for finger motion tracking involve conventional rigid solutions based on cameras[7] and inertial measurement units[8], and flexible solutions using stretchable and flexible materials based on the mechanisms of resistivity[9,10], capacitivity[11,12], and optical fiber[13,14], etc. To further lower the device power consumption, self-powered sensing mechanisms, including triboelectricity[15], piezoelectricity[16], thermoelectricity[17] and

[1]Department of Electrical & Computer Engineering, National University of Singapore, 4 Engineering Drive 3, Singapore 117583, Singapore. [2]Center for Intelligent Sensors and MEMS, National University of Singapore, 4 Engineering Drive 3, Singapore 117583, Singapore. [3]Singapore Institute of Manufacturing Technology and National University of Singapore (SIMTech-NUS) Joint Lab on Large-area Flexible Hybrid Electronics, National University of Singapore, 4 Engineering Drive 3, Singapore 117583, Singapore. [4]National University of Singapore Suzhou Research Institute (NUSRI), Suzhou Industrial Park, Suzhou 215123, China. [5]Printed Intelligent Device Group, Singapore Institute of Manufacturing Technology, Agency for Science, Technology and Research (A*STAR), Singapore 637662, Singapore. [6]NUS Graduate School-Integrative Sciences and Engineering Program (ISEP), National University of Singapore, Singapore 119077, Singapore. ✉ e-mail: elelc@nus.edu.sg

pyroelectricity[18], also reveal their unique advantages in developing the long-term sustainable portable system.

Due to the merits of diversified material choices, simple fabrication process, low cost and self-generated signals, sensing technology based on triboelectricity provides a new possibility to develop the highly flexible finger perception system with extremely low power consumption. Several glove-based advanced human-machine interfaces (HMIs) have been successfully achieved by using triboelectric nanogenerator (TENG) sensors for fine finger motion tracking[19,20]. A yarn-structural TENG strain sensor has been reported by Zhou et al. for gesture recognition[21]. He et al. developed a glove-based HMI with arch-shaped TENG textile sensors for intuitive control in cyberspace[22]. Though these studies have proved that TENG sensors can respond well to dynamic finger movements, their capability for static status or continuous motion sensing is limited due to the peak-like signals. To realize continuous sensing, TENGs with well-designed grating electrodes that can better quantify movements based on the generated peak numbers have been reported recently[23–25]. However, the size and spacing of the grating limit their resolution, which is difficult to reach a high level. Other solutions based on high-impedance readout circuits[26] or nanophotonic modulators[27] require complicated measurement systems, making them unsuitable for daily wearable/portable scenarios. Therefore, a more convenient approach to realize the continuous sensing function for TENG-based sensors compatible with mobile platforms is needed. Additionally, pyroelectric and thermoelectric materials are known to self-generate electrical signals based on varying temperatures and temperature differences, respectively[28–30], which have been frequently investigated as self-powered temperature sensors for HMIs[31]. Yang et al. developed a self-powered temperature sensor based on the single micro/nanowire pyroelectric nanogenerator whose resolution could be as low as 0.4 k at room temperature[32]. By fusing the thermoelectric, triboelectric and piezoresistive sensing mechanisms in one device, Yang et al. firstly proposed a self-powered multifunctional tactile sensor that can detect pressure and temperature, and identify material simultaneously[33]. The emerging polymer pyroelectric materials with high flexibility, e.g., poly-vinylidene fluoride (PVDF), etc., show the great potential to be integrated with TENG flexible materials to provide additional sensory information for fully self-powered multimodal sensing purposes[34]. Meanwhile, the emerging artificial intelligence (AI) based data analytics reveal the possibility of enriching the sensor functionalities to realize intelligent sensing[35–41]. With machine learning's strong feature extraction capability, subtle valuable features hidden in the complex signal outputs could be perceptible, and utilized to achieve advanced perceptions, e.g., gesture recognition[42–45], object identification[46,47], etc. Sundaram et al. developed a scalable tactile glove with assembled 548 piezoresistive sensors for high-accuracy grasped object recognition[48]. Kim et al. reported an electronic skin that can decode the complex finger motions enabled by rapid situation learning[49].

Wearable actuation systems are also indispensable parts to enhance the interactive experience in metaverse and have been developed with various mechanisms to enable haptic sensation[50]. There are actuators based on tendon driver[51], pneumatic[52], and electrostatic[53] mechanisms that can generate large-scale forces to provide kinesthetic feedback. Secondly, the electrotactile simulators[54] and the vibrotactile devices enabled by electromagnetics[55,56], electroactive polymer[57,58] and piezoelectrics[59] can simulate afferent nerves or mechanoreceptors in the skin for cutaneous stimuli. Compared with bulky kinesthetic actuation systems and other cutaneous stimulators requiring an external/high supply power, electromagnetic vibrators possess the merits of small size and low-driven voltage[60]. These features allow them to be integrated into wearable devices and simulate sensations of physical touch under the portable platform. Yu et al. reported a skin-integrated wireless haptic interface consisting of an electromagnetic vibrator array for VR interactions[55]. Jung et al. developed a wireless electromagnetic haptic interface to display vibro-tactile patterns across large areas of the skin[61]. Thirdly, the thermal sensation is also crucial to deliver a more realistic artificial feeling. For example, a few wearable devices based on Joule heating[62,63], thermoelectric effect[64,65] and electrocaloric effect[66] have been developed for thermo-haptic feedback. Among these technologies, metal- or carbon-based wire heaters exhibit the advantages of high flexibility/stretchability and loose operation conditions[67], enabling them to be easily embedded into daily wearable items and provide accurate and fast feedback on small areas of skin. Kim et al. reported a thermo-haptic glove enabled by a highly stretchable nanowire heater to simulate the feeling of heat[68]. Besides, to achieve a more immersive VR experience, a highly integrated wearable system with multimodal sensing and feedback functions that can provide diversified somatosensory perception and feedback simultaneously is needed[69]. Oh et al. developed a multimodal sensing and feedback glove integrating resistive strain sensors, vibrators, and thermo-haptic feedback units[4]. However, most current wearable solutions for VR sensing/feedback devices are glove-based with complicated structures and require large/external driven power. A highly integrated multifunctional ring-shaped manipulator with compact design, self-powered sensors and low-voltage driven feedback units that are fully compatible with the internet of things (IoT) portable platform for long-term usage purposes in VR application has not been achieved.

Herein, we propose augmented tactile-perception and haptic-feedback rings (ATH-Rings) for VR applications, which consist of TENG tactile sensors for continuous bending sensing, flexible pyroelectric sensors for temperature detecting, eccentric rotating mass (ERM) vibrators for vibro-haptic feedback, and nichrome (NiCr) metal wires for thermo-haptic feedback. All the sensors and haptic stimulators are integrated into the ATH-Rings with wires connected to a wireless IoT module as illustrated in Fig. 1. This minimalistic-designed ring with multimodal sensing and feedback functions shows the merits of a high level of integration and good portability compared to other state-of-art glove-based solutions listed in Supplementary Table 1. Moreover, the ATH-Ring uses self-powered sensing units and low-voltage driven feedback elements to achieve low power consumption. The proposed signal processing method based on voltage integration allows continuous finger motion detection based on TENGs. By leveraging the IoT module and machine learning (ML) analytics for signal collection and processing, advanced human-machine interactions, i.e., robotic collaborative operation, and sign-language translating, can be implemented for industrial automation and healthcare applications at low cost. Furthermore, by bringing in the haptic feedback functions enabled by vibration and thermal stimulators, augmented VR chat application with the interactive perception of the real world and the virtual world can be achieved using ATH-Rings, which gives the user a face-to-face like immersive experience and shows good development prospect for metaverse based social connections especially considering the strict social distance under the pandemic outbreak situation.

## Results

### System design and working mechanism

The structure of the ATH-Ring enabled system is shown in Fig. 1. The ATH-Ring unit on each finger achieves independent multimodal sensing and feedback (Fig. 1b). The 5 ATH-Rings are assembled with the electrical wires connected to the customized IoT module fixed on the rear area of the hand as illustrated in Fig. 1c. Polytetrafluoroethylene (PTFE) tubes are used as the wire package to strengthen the connections between different units. The customized IoT module is composed of a signal processing circuit, a wireless transmission unit, a microcontroller unit (MCU) with embedded analog-to-digital (ADC)

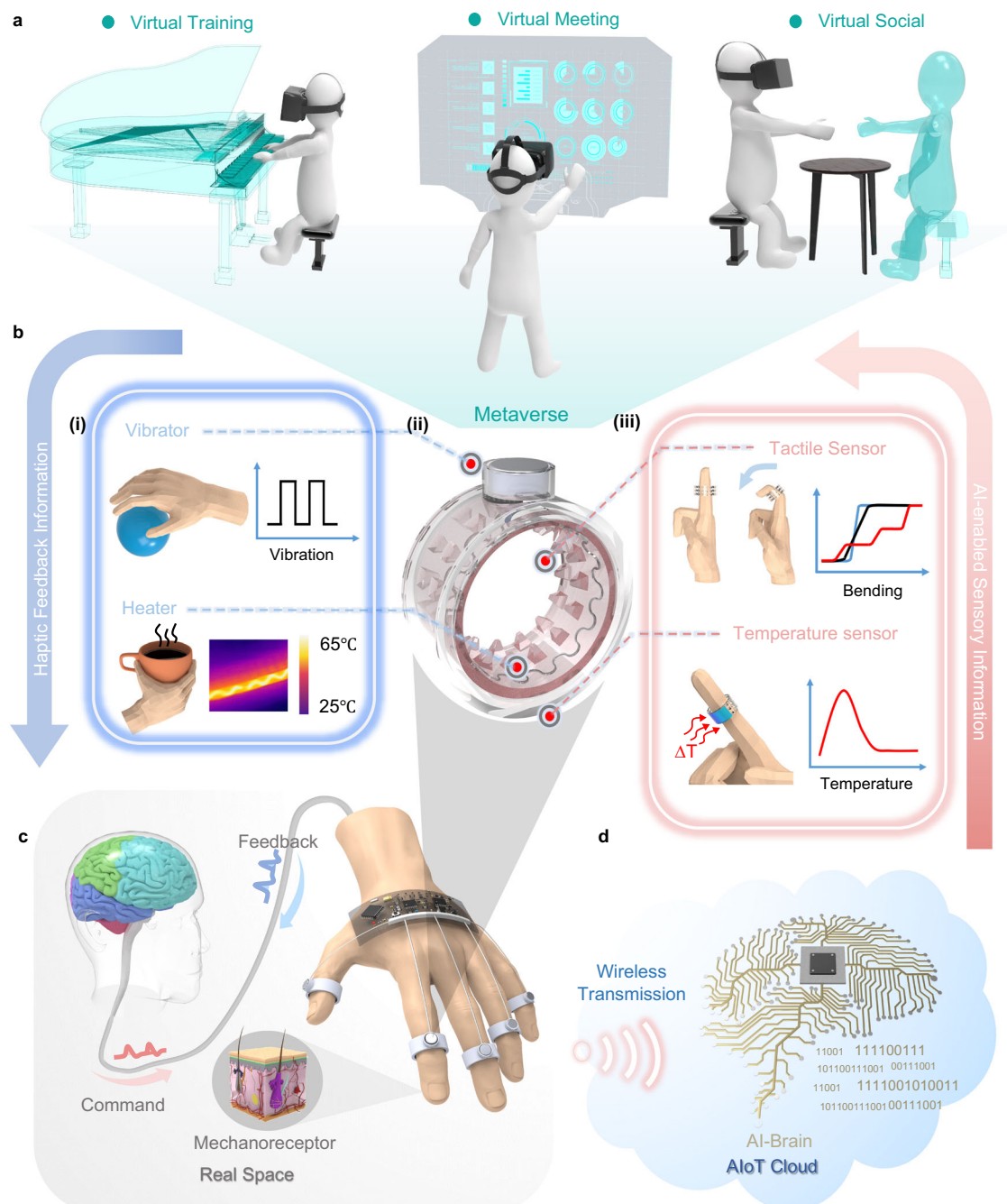

**Fig. 1 | The ATH-Ring based intelligent system with multimodal sensing and feedback functions. a** Illustration of the metaverse virtual space for augmented entertainment, education and social experience. **b** (i) The feedback functionality achieved by the integrated vibrator and heater. (ii) The detailed structure of the ATH-Ring. (iii) The sensing functionality achieved by the integrated tactile and temperature sensor. **c** Schematic of the biological neural network corresponding to the human finger sensation in real space. **d** The AIoT Cloud for wireless transmission and processing of the collected sensory information.

converters and pulse width modulation (PWM) pins to drive the haptic feedback units.

The illustration of the overall system, and the detailed structure of the different components can also be found in Supplementary Fig. 1. The TENG tactile sensor with the NiCr heater is mounted on the inner surface of the ring, which is in direct contact with the finger skin to precisely monitor the muscle swelling during finger bending as well as provide thermo-haptic feedback to the skin. The fabrication process of this unit is illustrated in Supplementary Fig. 1b and clearly stated in the Methods. The ERM vibrator is located at the top of each ring to deliver vibration to the entire finger, while the PVDF sensor is attached to the outer surface of the ring to measure the temperature of the object

being touched during grasping. Other components for connection or assembly purposes are all 3D printed with thermoplastic polyurethane (TPU) soft material as depicted in Supplementary Fig. 1d and the detailed printing parameters are provided in Supplementary Table 2.

The basic working mechanism of the TENG bending sensor is explained in Supplementary Note 1. In this sensor (Fig. 2a(i)), a layer of silicone rubber film with pyramid structures on one side is attached to the inner surface of a TPU ring serving as the negative triboelectric material due to its high electron affinity. The finger skin tends to lose surface electrons when contacting the silicone rubber film, thus acting as the positive triboelectrification layer. The aluminum film attached between the silicone rubber and the TPU ring acts as the output

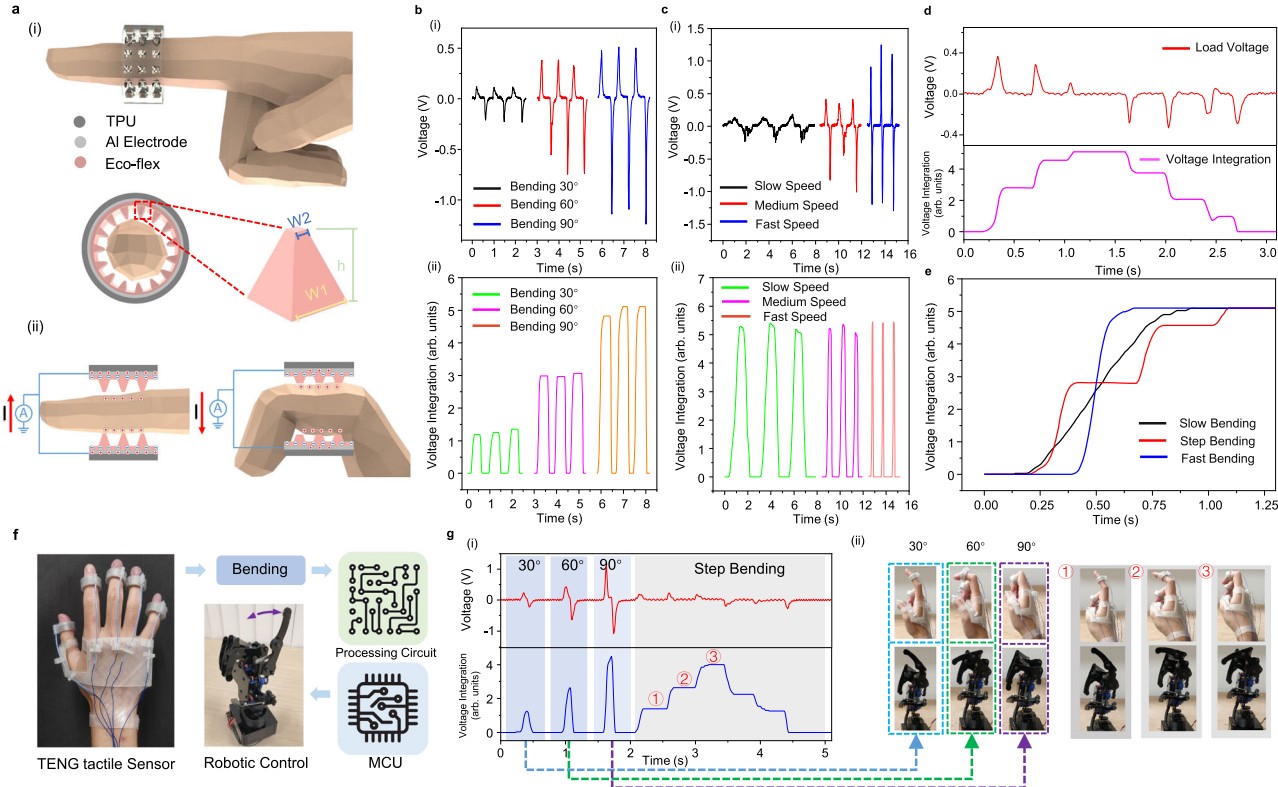

**Fig. 2 | Characterization of the ATH-Ring for continuous bending sensing. a** The (i) structure and (ii) working mechanism of the TENG tactile sensor. **b** The (i) load voltage and (ii) the corresponding integration value for bending with different angles and the same speed. **c** The (i) load voltage and (ii) the corresponding integration value for bending with different speeds and same angle (90°).

The verification of the tactile sensor for (**d**) step bending sensing and **e** continuous bending sensing. **f** The schematic of the robotic collaborative operation system. **g** The (i) outputs of the TENG tactile sensor for continuous robotic finger bending control with the (ii) illustrations show the corresponding human finger and robotic finger motions.

electrode of the sensor. The working mechanism of the pyroelectric temperature sensor is explained in Supplementary Note 2 and Supplementary Fig. 2. This kind of momentary temperature change induced output allows our ATH-Ring to project the temperature-related information of the grasped object in the real space into the VR space in real-time. In terms of the feedback units, both the ERM vibrators and the nichrome wire are directly driven by the IoT module to provide the vibro- and thermo-haptic feedback respectively. The detailed schematic shows all connections between the sensing/feedback units and the IoT modules in Supplementary Fig. 3a. The vibration amplitude and feedback temperature are both adjustable parameters via programming the PWM outputs of the MCU, so as to achieve adaptive feedback to truly reflect the variation in the stimuli in the virtual space. The illustration of the overall system including the power supply battery is shown in Supplementary Fig. 3b, c. All the functional units can be directly driven by a battery to achieve a wireless portable/wearable HMI under the IoT platform. This highly integrated multimodal sensing/feedback system shows a compact design and better portability when compared with most related works that need external/large driven power as listed in Supplementary Table 1.

From the system level, the sensory information of the finger bending or gripping is collected in real-time by the IoT module and wirelessly transmitted to the AI-enabled cloud server. In addition to real-time finger control and intuitive feedback, more advanced functions such as grasped object recognition are implemented via AI analytics to project diversified information from the real space to the metaverse virtual space. Such information could be utilized to reconstruct the grasped object in a virtual format that can be remotely felt by another user in the same metaverse virtual space through the simulated stimuli provided by the ATH-Ring, thus achieving an

interactive metaverse platform that provides users with a social experience with cross-space perception.

## Characterization of the ATH-Ring for continuous bending sensing

The detailed working process of the TENG tactile sensor is shown in Fig. 2a(ii). When the finger bends, the contact area between the finger skin and the silicone rubber pyramid structure will increase due to the muscle swelling, resulting in the electrical potential change in the output electrodes based on the triboelectrification on the contact surface. This potential variation further drives the electron flow and thus generates the outputs due to electrostatic induction. The output optimization of the TENG tactile sensor in terms of different pyramid sizes/diameters and ring diameters is illustrated in Supplementary Fig. 4 and Supplementary Note 3.

To enable the TENG sensor with continuous motion sensing capability for ATH-Rings, the signal processing method based on voltage integration is proposed (see Methods). The outputs of the TENG tactile sensor for measuring different bending degrees are shown in Fig. 2b. When the bending speed is the same, the greater the bending angle of the finger, the larger the output voltage and voltage integration value are. Actually, most of the TENG-based methods quantify the bending angle based on the output voltage amplitude[22,44,69]. However, as shown in Fig. 2c(i), when bending at the same angle, the bending speed also affects the amplitude of the output voltage, resulting in the inability to accurately measure the bending angle. However, the voltage integration value, as shown in Fig. 2c(ii), is virtually unaffected by the bending speed, which can be used as a reliable reference to measure the bending angle of the finger. Figure 2d, e show the capability of the voltage integration method for continuous detection. In Fig. 2d,

the load voltage can only reflect the angle of each step bending, while the voltage integration value curve can record the continuous angle change during the whole bending process. By comparing the plots in Fig. 2e, bending speed information is further extracted from the slope of the voltage integration curves to enable more comprehensive strain/tactile sensing. Supplementary Fig. 5a shows that the changing trend of the transferred charge with the bending angle is highly similar to that of the integration value with the bending angle. It demonstrates the feasibility of using the voltage integration value to represent the transferred charge for continuous bending detection. Besides, when the bending angle is increased from 20° to 30° in the 1° interval as shown in Supplementary Fig. 5b, the variation of the bending degree can still be well distinguished, showing the strong perceiving ability of our developed TENG tactile sensor. The high resolution of the sensor is also verified in Supplementary Movie 1. The detailed advantages of voltage integration over other signal processing methods for TENG-based strain sensors are highlighted in Supplementary Note 4 and Supplementary Table 3.

By integrating five TENG tactile sensors with 3D-printed soft connectors and using a custom IoT module for signal collection and transmission, an advanced manipulator is successfully achieved to implement the real-time and continuous robotic collaborative operation as shown in Fig. 2f. Figure 2g shows the sensor outputs of the continuous bending control of a robotic finger. In the left part (blue) of Fig. 2g(i), the robotic finger is controlled to directly bend at a large angle (30°, 60°, 90°) and release to 0°, where the voltage integration value rises and drops sharply to reflect these instantaneous motions, showing the fast response sensing capability of the control interface. The right part (gray) of Fig. 2g(i) demonstrates the continuous control capability of the system, where the robotic finger is controlled to bend at a small angle each time and held for a while until bending to the 90°. The detailed operation process can be found in Supplementary Movie 2. In addition, the multi-finger control is also achieved based on the independent outputs of sensors on each finger to realize complex gesture manipulation as demonstrated in Supplementary Fig. 6. The corresponding fingers of the robotic hand will bend according to the bending fingers of the actual hand, and there is nearly no interference in this multi-channel signal collection process, proving the reliability of this collaborative operation system for complex gesture control for practical usage.

## ML-enabled gesture/sign language recognition system

Although many works have shown the possibility of using TENG sensors to implement high-accuracy gesture/sign language perception systems with AI analytics[21,45], almost all of them utilized the pulse-like output signals as the input of the learning architecture. The impact of pulsed and continuous signals on interpretation performance has not been discussed so far.

Here we selected 14 American sign language gestures as illustrated in Fig. 3a to test the recognition performance based on the pulse-like signals (Fig. 3b(i)) and voltage integration signals (Fig. 3c(i)), respectively. A normalization process (see Supplementary Note 5 and Supplementary Fig. 7) is adopted during the dataset establishment to avoid the sensor output difference caused by various finger sizes in one hand or between individuals, ensuring the generalization ability of the system in practical applications. The bending angle of fingers for each gesture can be reflected both by the peak values in the pulse-like spectrum or the stable values in the voltage integration spectrum. Each gesture category contains 120 samples, where 80 samples were used for training and 40 samples were used for testing. By using the principal component analysis (PCA) for feature reduction, the preliminary classification results were visualized in Fig. 3b(ii) and Fig. 3c(ii). It is clear that the aggregation effect based on the voltage integration signal is better. While for the pulse-like signals, many categories are mixed together, indicating that the 3 features extracted from the

pulse-like output spectrum used to distinguish different categories are not so effective. A similar result could be achieved when adding a supporting vector machine (SVM) classifier for further identification. As shown in Supplementary Fig. 8a and the confusion maps in Fig. 3d and e, the highest accuracy for the voltage integration based dataset reaches 99.821% based on 8 principal components, while the highest accuracy for the pulse-like signal based dataset can only reach 97.143%. Though the voltage integration method appears to be just a further data processing of the load voltage, the feature differences between different gestures are more obvious after such processing, which helps the machine learning model to better extract and interpret. Between, human-induced disturbances, e.g., the influence of the difference in bending speed on the voltage amplitude (Fig. 2c), are also diminished in the integration signal, which may result in the higher accuracy. When compared to other state-of-art wearable gesture recognition systems listed in Supplementary Table 4, our work can achieve comparable performance even with a smaller number of sensor nodes. Between, we have also done the durability test of the TENG tactile sensor for long-term use. As shown in Supplementary Fig. 9, after thousand cycles of utilization, the output of the sensor did not decay. Because the recognition performance of our system is based on the final integration of the voltage output, as long as the output does not change much, the accuracy of the identification will not be affected.

In addition to the better recognition performance for single gestures, compared to the abovementioned previous TENG-based works, the voltage-integration approach also shows advantages in continuous sign language interpretation, where the action of making a specific gesture will be influenced by the gesture of the previous moment. For example, if we make a single gesture of "2", we need to bend three fingers: thumb, ring finger and pinky finger. However, suppose we do the gesture of "1" first, then "2". In that case, we just need to release our middle finger, which is quite different from the motion of the single gesture of "2". This effect will result in the difference in the pulse-like signal between the single gesture, and the same gesture in a continuous sentence as shown in Fig. 3f(i). However, since the integration signal is continuous, it can be maintained to reflect the state of each finger, and the effect of the previous gesture can also be ignored as long as we select the final stable value as the gesture signal as indicated in Fig. 3f(ii), meaning that the signal of single gestures can be directly used in the sentence. This feature helps us greatly save the time and labor cost of collecting gestures, and improves the universality of the data set. Based on this, we rebuild a data set that only contains the final stable values of the voltage integration signals and then retrained. The testing result is shown in Supplementary Fig. 8b, where the accuracy of 99.821% is maintained, proving the effectiveness of such a method to maintain performance and provide convenience in continuous sign language interpretation.

## Augmented haptic feedback system compatible with IoT platform

Besides the advanced sensing capability, the haptic feedback functionality of the VR wearable device is also indispensable to give users a simulated sensation to enhance the interactive experience in the virtual environment.

A tactile feedback system with low driven power is realized by integrating the ERM vibrators onto the ATH-Ring as illustrated in Fig. 4a(i). Instead of being put inside the ring and close to the skin, the vibrator is located at the top of each ring to deliver vibration to the entire finger considering the limited sensing area of the TENG tactile sensor and integrity of the whole device. Although the vibration is slightly attenuated in this case, it is still strong enough to provide a varying vibration intensity and noticeable difference in haptic information for users to perceive. To calibrate the vibration amplitude of the vibrator, a piezoelectric vibration sensor is utilized and the

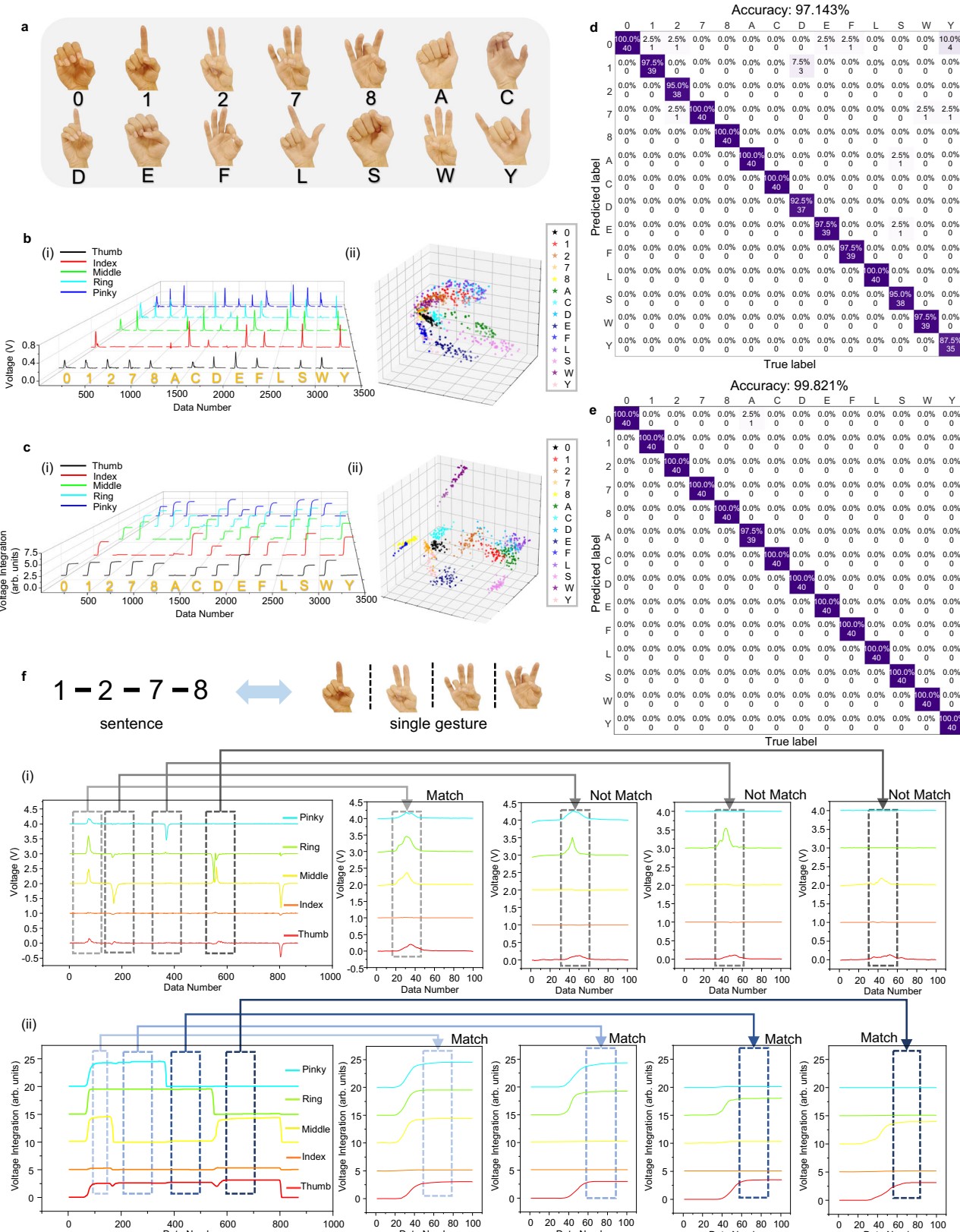

**Fig. 3 | ML-enabled gesture/sign language recognition system enabled by the ATH-Ring. a** Illustration of the 14 American sign language gestures for further recognition. **b** The (i) pulse-like signals (ii) and the corresponding PCA clustered results for 14 gestures. **c** The (i) voltage integration signals (ii) and the corresponding PCA clustered results for 14 gestures. The corresponding confusion map for (**d**) pulse-like signal dataset and **e** voltage integration signal dataset. **f** The (i) pulse-like signal spectrum and (ii) voltage integration signal spectrum for a continuous sign language sentence: 1-2-7-8.

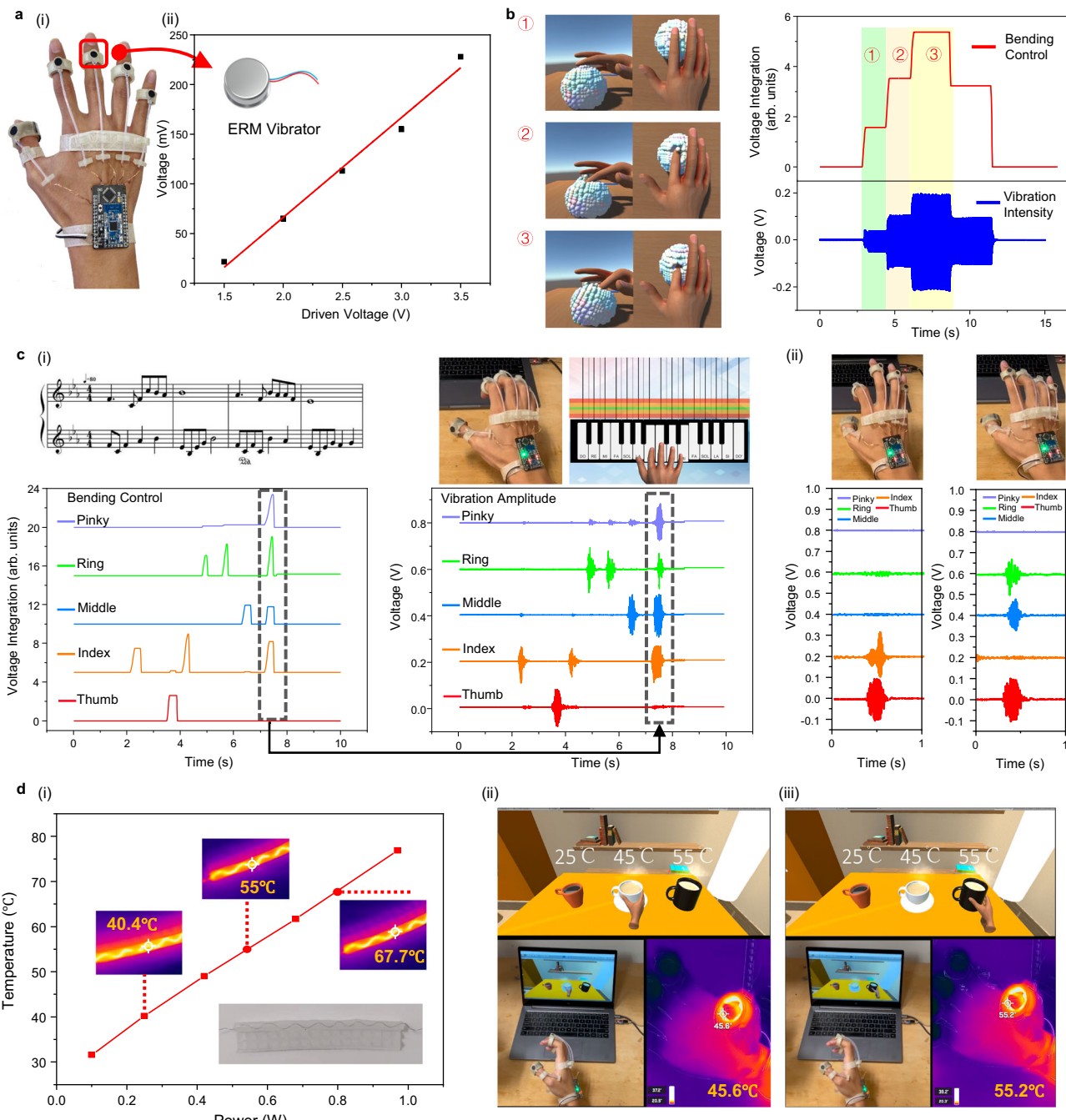

**Fig. 4 | Augmented multimodal haptic feedback system compatible with IoT platform. a** (i) Illustration of the ATH-Ring based HMI with embedded ERM vibrators. (ii) The relationship between the vibration intensity collected by a PZT vibration sensor and the driven voltage of the vibrator. Demonstration of the vibro-haptic feedback function via **b** squeezing a soft ball with one finger and **c** playing a virtual piano with multiple fingers, with corresponding control and real-time vibration signals. **d** Thermo-haptic feedback enabled by the NiCr metal wire heater. (i) The stable temperature of the NiCr heater enabled temperature feedback unit at different driven power. Demonstration of thermo-haptic feedback when grasping a hot coffee with the temperature of (ii) 45 °C and (iii) 55 °C in the virtual space.

measured result is plotted in Fig. 4a(ii). The approximately linear relationship between the driven voltage, sensor output, and the actual vibration amplitude (also see Supplementary Fig. 10a) can help provide adjustable vibration intensity in the actual applications. Besides, as plotted in Supplementary Fig. 10b, the ERM vibrator changes both the vibration frequency (130–230 Hz) and amplitude under different supply voltages. This feature can help it introduces a stronger change of feeling for users[61,70] when compared to other vibrators, i.e., linear resonance actuator, piezoelectric actuator and voice coil actuator[55], that operate at a fixed resonant frequency. In Fig. 4b and

Supplementary Movie 3, a virtual hand is controlled to squeeze a soft ball in the virtual world based on the actual bending angle of our finger detected by the TENG tactile sensor. The feedback intensity during this process has three stages. The first stage is that when the finger just touches the soft ball without applying an external force, the amplitude of the vibrator is very small. In the second and third stages, the soft ball is gradually deformed by the human virtual finger. The driven voltage will increase accordingly, so the intensity of the vibration will increase to provide stronger vibro-haptic feedback to simulate the actual feeling of squeezing. Here, the vibration intensity is related to the hardness

of the virtual object, and the clearer control logic is provided in Supplementary Fig. 11. For pressing the soft object mentioned above, the vibration intensity will increase with the degree of deformation of the soft object, and will reach the maximum when the soft object can no longer be squeezed. However, for pressing rigid objects which are undeformable, the vibration intensity reaches its maximum at the moment of contacting the surface of the object. In addition, the slope of the curve of vibration intensity versus bending angle can be adjusted according to the stiffness of the object, where a larger slope means greater stiffness. Through this kind of control logic and feedback system, the haptic perception of objects under different stiffness could be well mimicked. Additionally, the simultaneous multi-finger control and haptic feedback can also be achieved via a VR piano training demonstration in Fig. 4c (also see Supplementary Movie 4). The ring on each finger can independently monitor the finger motion for real-time control, and provide specific vibration feedback after receiving the collision signal generated in the virtual space, showing the potential for future virtual educational training applications.

Besides vibro-related feedback, thermo-haptic feedback is also an important function to provide users with a more comprehensive perception of the object. Here, we embed the NiCr metal wire heater into the TENG tactile sensor as a thermal feedback unit as illustrated in Fig. 4d(i), whose temperature could be easily heated up to a high temperature with low power. The relationship between the driven power and the final maintained temperature is also plotted. The response time of the heater corresponding to different driven power is shown in Supplementary Fig. 12a. A higher driving power achieves a higher stable temperature, but at the same time, it also requires a longer response time. The relatively long response time is caused by the silicone encapsulation effect, but is still acceptable when compared with other related works[4]. For application scenarios requiring faster response time, another driven strategy could be utilized by first using high power to reach the desired temperature and then lowering the power to maintain the temperature (Supplementary Fig. 12b). However, higher power and longer response time are needed for the heating interface to reach the same temperature in practical applications when the device is in contact with human skin (see Supplementary Fig. 13), where 61.9 °C is achieved within 9.4 s under the maximum output power (-3.5 W) of the IoT platform, which could still meet the requirement of most scenarios considering the suitable perceiving temperature range of the human body. In order to achieve optimized power consumption of the overall system (see Supplementary Note 6) and considering the necessity of the thumb involved in the grasping tasks, we only add one thermo-haptic feedback component to the thumb ring to reduce the power consumption and the complexity of the entire system. As illustrated in Fig. 4d(ii-iii), a virtual space with coffee cups of different temperatures is established. A virtual hand can be controlled in real-time by the TENG sensor to grasp the coffee cup first. Then the embedded NiCr heater will be heated up to a specific temperature according to the predefined temperature value of the selected coffee cup. The final stable temperature captured by the infrared camera is almost equal to the preset temperature of the corresponding coffee cup in the virtual space, verifying the temperature feedback capability of the proposed system (also see Supplementary Movie 5). Furthermore, heat flux measurement information is also useful to access the temperature sensation. As plotted in Supplementary Fig. 14a, the heat flux increases with the applied power, and the heat can be easily transferred to the skin under certain supply power. Besides, the approximately linear relationship between the heat flux and the heater temperature (Supplementary Fig. 14b) enables us to control the heat flux simply by controlling the temperature of the heater, which could also be used to mimic various thermal feelings in specific applications, e.g., differentiating objects with different thermal conductivity at the same temperature[68], etc. However, limitations still exist with the current thermo-haptic unit, e.g., longer cooling time

and inability to provide the cold feeling when compared to devices based on thermoelectric active cooling[64,65]. Moreover, relatively large power consumption happens when multiple fingers are equipped with thermal heaters. Hence, further improvement to the design of the heater will be necessary in the future.

Using this improved glove for grasping perception in VR space, accurate thermo- and vibro-haptic feedback could be realized simultaneously. Supplementary Note 7 and Supplementary Figs. 15–16 also demonstrate that there is nearly no mutual interference between the haptic feedback units and tactile sensor, revealing the possibility of building a fully portable multifunctional sensing and feedback HMI for metaverse applications.

## Multimodal sensing and feedback platform for metaverse

The current metaverse-based virtual social platform is still limited by conventional interactive media, i.e., visual or voice. To further enhance the interactive experience, the human sensation from other body parts is also essential for a more comprehensive perception and feedback experience. Based on this consideration, we proposed an augmented VR chat platform enabled by the ATH-Rings as illustrated in Fig. 5a, where the two users can achieve the cross-space perception and sensation attributed to the multimodal sensing and feedback capabilities brought by the highly integrated system. As depicted in the system schematic, first, user 1 grasps an object in real space. Then the corresponding shape and temperature information could be collected by the TENG tactile and PVDF temperature sensor, respectively. The customized machine learning analytic performs the object recognition in the cloud server based on the collected sensory information. Such recognition results can be projected to a virtual space in the metaverse that other users can access. Meanwhile, user 2 can perform the real-time finger control to touch the projected virtual object generated from user 1 side, by which the shape and temperature related sensory information are feedback to the user 2 side and used to drive the vibrators and heaters to simulate the real touch sensation in the actual space of user 2.

To realize such a system, the grasped object recognition function is needed. The gesture recognition ability of ATH-Rings has been demonstrated in the above section on machine learning, and the object recognition could also be realized based on the variation of sensor outputs in finger motion when grasping different objects. To verify the object recognition capability of the developed ATH-Rings, we collect the gripping data of 5 blocks, i.e., cube, cylinder, tri-pyramid, big ball and small ball, with different sizes and shapes as illustrated in Fig. 5b. The data set is built via repeating grasping of each object by 120 times and collecting the voltage integration signal. The data length for each channel is 300, so the total features of each sample are 1500, considering the 5 ATH-Ring units on one hand. 80 samples out of 120 samples of each object in the data set are used for training, and the remaining 40 samples are used for testing. The recognition result through a customized SVM classifier is shown in the confusion map and a relatively high recognition accuracy of 94% is achieved. Considering the similar gestures of gripping motions, it is acceptable and shows the feasibility of using the finger bending information to reflect the grasped object shape. To further investigate the feasibility of this system for actual application, another data set containing 8 common daily items is built and illustrated in Supplementary Fig. 17, where the recognition accuracy is higher than 96% because of greater variability in volume and shape compared with the abovementioned five blocks.

Besides the shape-related information from the TENG sensors, the temperature sensing function is also important to bring in more comprehensive information in order to enhance the recognition capability. Here, a PVDF temperature sensor with the advantage of self-generated output and high flexibility is utilized. It is integrated with the TENG tactile sensor to form a fully self-powered sensing system. The

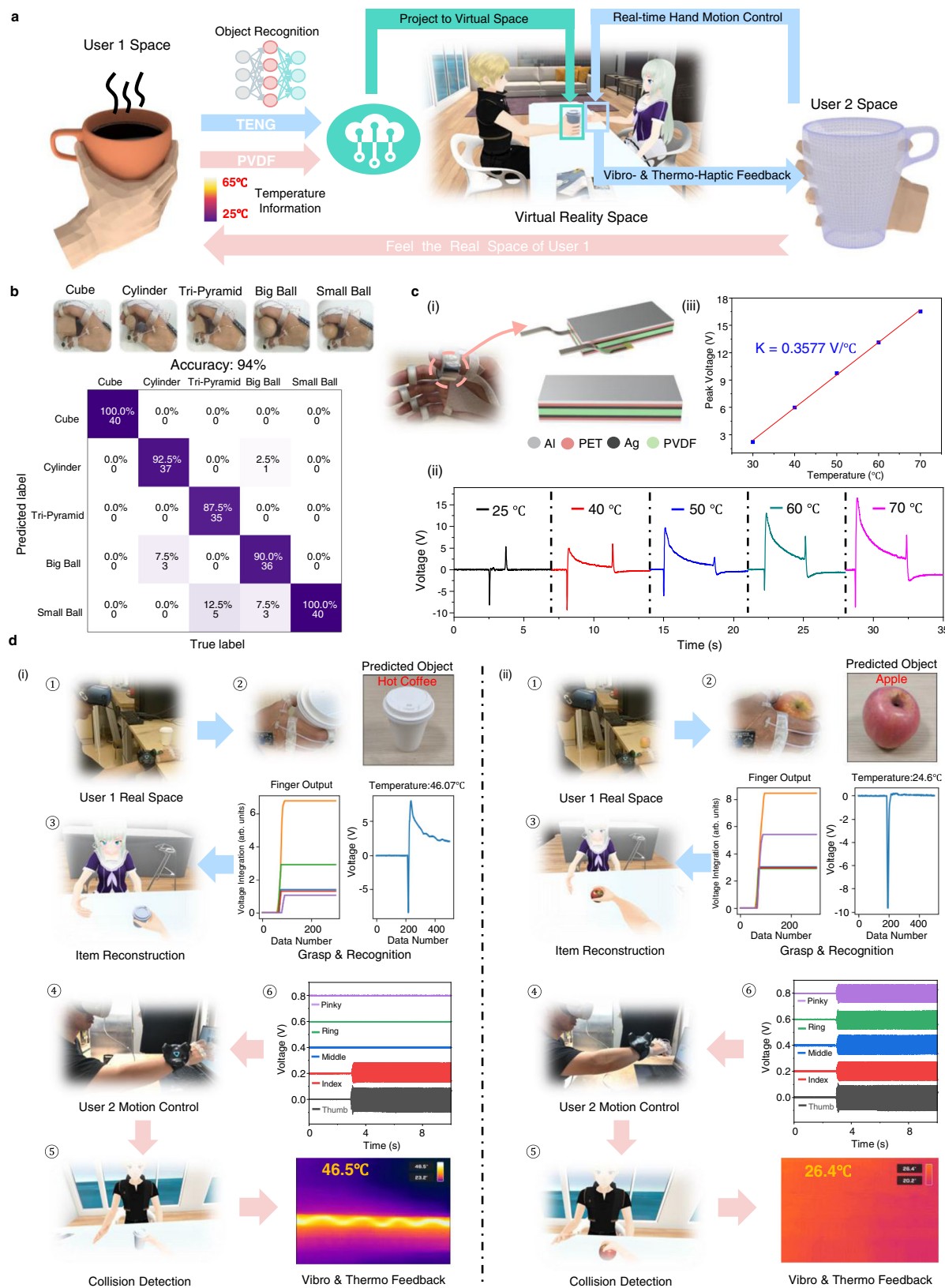

**Fig. 5 | Multimodal sensing and feedback platform for metaverse. a** System architecture of the ATH-Ring enabled interactive metaverse platform to provide users with cross-space sensation. **b** The illustration and the corresponding confusion map of 5 blocks with different sizes and shapes for verifying the grasped object recognition capability of the ATH-Ring enabled HMI. **c** The (i) detailed structure of the PVDF temperature sensor, the (ii) output waveform and the (iii) corresponding calibration result of the PVDF sensor when contacting objects with different temperatures. **d** The real-time system operation process of the metaverse platform with the testing object of (i) a cup of hot coffee and (ii) a room temperature apple.

detailed structure is illustrated in Fig. 5c(i), where a layer of poled PVDF film is covered by the silver output electrodes on both sides. The polyethylene terephthalate (PET) material is utilized for the sensor package, with additional ground electrodes attached to the outer surface to reduce the triboelectricity-induced noise as well as maintain good thermal conductivity. Figure 5c(ii) shows the output of the PVDF film when in contact with objects under different temperatures at the same contact force. When the temperature of the contacted object is higher than the original temperature of the PVDF sensor, a positive voltage peak is generated due to the greater spread of the electric dipoles on their respective alignment axes in PVDF. Besides, since PVDF material has piezoelectric properties, it also produces a pressure-related signal when contacting objects. The pressure-induced signal is a narrow negative peak that appears before the temperature-induced positive peak as shown in Fig. 5c(ii). The influence of pressure on temperature output is also investigated and shown in Supplementary Fig. 18. When the temperature of the touched object is fixed at 45 °C, the temperature-induced output increases slightly from 7.53 V to 8.01 V when the pressure increases from 10 kPa to 40 kPa due to the larger contact area under larger pressure, and saturates when the pressure is larger than 40 kPa. According to the calibration result shown in Fig. 5c(iii), the error due to the pressure difference in this process is about 1.34 °C, which is acceptable in practical applications. Figure 5c(iii) also depicts that the pyroelectric peak value is approximately linear in the temperature range of 30 °C to 70 °C and increases with the raised temperature. Hence, it can serve as a reliable reference for object temperature perception.

The operation details of the established system are shown in Fig. 5d and Supplementary Movie 6. It is worth noting that, since the developed ATH-Ring is designed for motion capture and feedback for fingers, the positioning of the entire hand in real space is achieved with the help of the HTC tracking system known as "Lighthouse". The specific working principle of this system can be found in Supplementary Fig. 19 and Supplementary Note 8. In Fig. 5d(i), a cup of hot coffee is firstly grasped by user 1 in the real space. Then based on the collected 5-channel TENG sensor output and PVDF sensor output, the shape information of the grasped object could be extracted by the machine learning analytic and fused with the temperature information to reconstruct the corresponding object in the virtual space, which is also visible to user 2 in a remote real space. After user 1 puts the hot coffee in both the real and virtual space, user 2 can control the virtual hand in real time to grasp the reconstructed virtual object. Here, user 2 uses the thumb and index finger to grasp the hot coffee. Upon detecting the collision signal in the virtual space, the vibrators and heater will start to operate to provide vibro- and thermo-haptic feedback. The vibration intensity and the feedback temperature could also be monitored and are visualized in Fig. 5d(i), which corresponds to the contact finger channel (thumb and index) and shows high similarity to the temperature value obtained by the infrared sensor from the user 1 side, verifying the multimodal sensing and feedback capability of the proposed system. As a result, the cross-space perception and sensation function could be achieved to help the user to feel the actual space of others. Similarly, the corresponding sensing and feedback result of grasping a room-temperature apple is shown in Fig. 5d(ii), and the obtained result further demonstrates the ability of the ATH-Ring to perceive different objects and provide diversified feedback. The good power efficiency (see Supplementary Note 6) and relatively fast response (see Supplementary Note 9) of the overall system also ensure a good user experience. Moving forward, the developed interactive perception of the real world and the virtual world enabled by the ATH-Ring can give people a face-to-face like immersive chat experience, showing good development prospects for metaverse based social connections, especially considering the strict social distance under the pandemic outbreak situation.

## Discussion

To enable somatosensory sensation for more immersive VR and metaverse applications, we develop a highly integrated ATH-Ring with multimodal sensing (tactile and temperature sensing) and feedback (vibro- and thermo-haptic feedback) capabilities. All the functionalities are implemented on a minimalistic designed ring and driven by a custom IoT module, demonstrating higher integration and portability than other similar works. Besides, the self-powered sensing features of the TENG and PVDF sensors could further reduce the power consumption of the whole system to enable a long-sustainable wearable manipulator under the IoT framework. The proposed novel signal processing method based on the voltage integration provides the possibility to realize continuous motion detection with TENG sensors on mobile platforms, and also contributes to higher-accuracy gesture recognition, i.e., 99.821% for 14 sign language gestures, when leveraging the ML data analytics. The voltage-integration approach also shows advantages in continuous sign language interpretation by eliminating the discrepancy of signals between the single gesture and the corresponding gesture in consecutive sentences. Furthermore, by utilizing the sensor signal to trigger the integrated vibrators and heaters, adjustable vibro- and thermo-haptic feedback is achieved to simulate the sensation of touching objects in the virtual space. Based on this AI-enhanced multimodal sensing and feedback system, an interactive metaverse platform that provides users with cross-space perception capability is successfully achieved, where the object in the real space of one user could be recognized and reconstructed into the virtual format, and remotely felt in real time by another user in the same metaverse virtual space through the simulated stimuli and the feedback, giving people a face-to-face like immersive virtual social experience. As a future prospect under the metaverse infrastructure, by fusing the VR display technology with such multimodal somatosensory sensation for full-body perception and feedback, a smart virtual society could be built to enable intelligent and interactive social, education, entertainment, and healthcare, etc.

## Methods

### Fabrication of the TENG tactile sensor with NiCr metal wire heater

A 3D printed mold was prepared with pyramid grooves for fabricating the silicone-based tribo-layer. The mixture of solution A and B of Eco-flex (model:00-50) was poured into the mold, and cured at room temperature for 60 min to form the silicone rubber film. After this, a NiCr metal wire was attached to the area of the silicone rubber surface without the pyramids, then covered by an additional mixture of Eco-flex for packaging. After the second curing, this sensing and feedback unit was completed and ready for practical usage.

### Fabrication of soft components for connection and assembly

Components for connection or assembly purposes are all 3D printed with TPU 85A soft material and the detailed printing parameters could be found in Supplementary Table 2.

### Characterization of the TENG/PVDF sensor

The signal outputs in the characterization of the TENG/PVDF sensor were measured by an oscilloscope (DSOX3034A, Agilent) using a high impedance probe of 100 MΩ. The transfer charge was conducted by an electrometer (Model 6514, Keithley) and the signals were displayed and recorded by the oscilloscope. Analog voltage signals generated in TENG/PVDF sensors and the voltage integration signals for IoT applications were collected by the customized hardware circuit consisting of an ADC, an MCU and a wireless transmission module.

### Signal processing method based on the voltage integration for TENG tactile sensor

Based on the triboelectric theory, the transferred charge can be seen as a continuous parameter for TENG sensors to reflect the whole stimuli/movement, which can only be achieved by high-impedance measurement instruments. The calculation formula for the charge is $Q = I \bullet t = \int_0^t i(t)dt$, where $i(t)$ represents the instantaneous current flow through the load. Based on the formula: $v(t)_{load} = i(t) \bullet R$, we know $Q = \int_0^t i(t)dt \propto \int_0^t v(t)_{load}dt$, where the integration value of the load voltage is proportional to the transferred charge, which can also be utilized to represent the continuous movement/stimuli. More importantly, the integral value of the load voltage can be directly obtained by the IoT signal acquisition module through a certain integral algorithm, which is suitable for wearable/mobile scenarios.

### Characterization of thermo-haptic feedback

The heating temperature of the nichrome heater in thermo-haptic feedback related demonstration was measured by an infrared (IR) camera (FLIR One Pro). The response time profile of the heater when placed on a TPU substrate with another side exposed to the air was collected by the IR camera. The response time profile of the heater when placed on a TPU substrate with another side in contact with the skin was measured by a 100k Ohm thermistor sandwiched between the skin and the heating surface. The heat flux information was achieved by using a commercial heat flux sensor (FluxTeq) with the heater sandwiched between a TPU substrate and a heat flux sensor under a finger press.

### Characterization of vibro-haptic feedback

The vibration amplitude of the vibrators was calibrated by a piezoelectric vibration sensor, and visualized by an oscilloscope (DSOX3034A, Agilent). The actual vibration displacement and frequency were collected by fixing a vibrator on a TPU ring and measuring via a laser vibrometer (VIB-A-510, Polytec).

### Machine learning enabled by PCA and SVM

Principal component analysis (PCA) is often used to reduce the dimensionality of each data, while preserving the features that best reflect the variability of the data, in order to better distinguish data from different categories. Here we perform the analysis via the PCA module available in Scikit-learn library in Python 3.9 environment. The first three principal components were used to display 3D scatter plots of the features. After the dimensionality reduction process via PCA, we used a support vector machine (SVM) classifier for further classification. The SVM classifier is also available in Scikit-learn library. In both the gesture recognition and object recognition data analysis, we trained the SVM classifier with the linear kernel and set C parameter (penalty parameter of the error term) as 1.0.

## Data availability

All technical details for producing the figures are enclosed in Methods and Supplementary Information. Data are available from the corresponding author C.L. upon reasonable request.

## Code availability

The codes that support the findings of this study are available from the corresponding author C.L. upon reasonable request.

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

## Acknowledgements

This work was supported by Advanced Research and Technology Innovation Center (ARTIC) under the research grant of A-0005947-20-00 at National University of Singapore, Singapore; National Key Research and Development Program of China (Grant No. 2019YFB2004800, Project No. R-2020-S-002) at NUSRI, Suzhou, China; "Intelligent monitoring system based on smart wearable sensors and

artificial technology for the treatment of adolescent idiopathic scoliosis", the "Smart sensors and artificial intelligence (AI) for health" seed grant (Project No. A-0005180-16-00) at NUS Institute for Health Innovation & Technology (NUS iHealthtech), Singapore; the Collaborative Research Project under the SIMTech-NUS Joint Laboratory, "SIMTech-NUS Joint Lab on Large-area Flexible Hybrid Electronics", Singapore.

## Author contributions

Z.S. and C.L. conceived the idea. Z.S., M.Z. and C.L. planned the experiments. Z.S. designed and completed the hardware, and performed the experiments. Z.S. took all the photos shown in figures. Z.S. wrote the control programs and algorithms for demonstration. Z.S. and M.Z. took the demonstration movie. Z.S. contributed to the data analysis and drafted the manuscript. Z.S., M.Z., X.S. and C.L. edited the manuscript.

## Competing interests

The authors declare the following competing interests: Z.S., M.Z., and C.L. are inventors on the United States provisional patent application (application no. 63/352,712) submitted by National University of Singapore that covers the augmented tactile-perception and haptic-feedback rings and metaverse-based virtual social platform. The remaining authors declare no competing interests.
