## [Peer Review File · Nature Communications]

REVIEWER COMMENTS

Reviewer #1 (Remarks to the Author):

The authors propose a multimodal sensing and feedback ring for metaverse applications, which contains TENG- and PyENG-based self-powered sensors as tactile and temperature perception units, and vibrators and nichrome heaters for vibro- and thermo-haptic feedback. All these functionalities could be achieved on minimalistic designed rings and directly driven on wearable/portable platforms, showing a high level of integration compared to other glove-based perception/feedback related solutions. In contrast to the recent research trends in self-powered sensor enabled finger bending monitoring or gesture recognition system (Nat. Electron. 3, 571–578 (2020); Nat. Commun. 12, 5378 (2021)), the newly proposed voltage-integration approach not only makes the TENG sensors capable of monitoring continuous finger motion on mobile platforms, but also show the advantages in continuous sign language interpretation. Besides, a novel interactive virtual chat platform is also successfully demonstrated, to give users an immersive social experience for future metaverse based applications. I think this work do very comprehensive investigations on multimodal perception and feedback system development and the system integration for advanced human-machine interactions. Hence, I recommend the minor revision with the following comments:

1. In Fig. 2, the authors propose to use voltage integral signal from TENG sensors to measure the finger bending angles. Compared to other TENG-based related works, may the authors comment the detailed advantages of this kind of sensing method in a small table to help readers better understand the scientific value of this approach? Can the author comment on what scenarios are suitable for using voltage integral signal or original voltage signal?
2. The machine learning enabled gesture/object recognition function relies on the outputs of the TENG tactile sensors. How is the stability of the sensors and the integrated sensory system? After cycles of utilization, is the accuracy still be maintained? Another table comparing the recognition performance of this work and prior similar research is suggested to further highlight the strengths of this work.
3. The illustrations of the entire device/system shown in Figure 4a and Supplementary Figure 1 appear to differ in the appearance of the TPU connectors. If all the main functionalities are integrated in rings, and “other components just for connection or assembly purposes”, it's better to make it consistent throughout the whole manuscript.
4. In Fig. 4b, the authors show the vibration intensity variation while grasping a soft virtual object. However, in the main text, the authors claim that “Through this kind of control logic and feedback system, the haptic perception of objects under different hardness could be well mimicked.”, may the authors also provide the real-time vibration intensity of grasping rigid virtual objects and compare it with that of the soft one to make the statement more convinced.

5. For the actual application as interactive tools in combination of haptic feedback and sensation, feedback and perception often happen at the same time, and some mutual interferences might be unavoidable, e.g., the vibration of the vibrators will affect the output of the tactile sensor, the temperature change brought by the heater will change the working environment of the tactile sensor. So in this device, do these interferences exist? If yes, how to avoid such effects and make them work simultaneously?

6. What is the response time of the metaverse-based interactive system shown in Fig. 5 (including the sensor/actuator response time and wireless transmission time), i.e., the time from when the command is sent from one user side to when the haptic actuator starts to work from another user side.

7. Since the sensor is directly contacted with skin for triboelectrification. Will the pyramid structure cause uncomfortable feeling to user? How about the signal stability for different user with different finger size?

8. They need to add the description about the "It is the first time that Yang et al. have utilized one tactile sensor to realize the pressure, temperature and material identification. [Wang et al., Sci. Adv. 2020; 6 : eabb9083]" in the first part of the manuscript.

9. The pyroelectric nanogenerator and sensor's related papers should be cited and described :

[1] DOI: 10.1021/NL3003039

[2] DOI: 10.1002/ADMA.201201414

[3] DOI: 10.1021/NL303755M

[4] DOI: 10.1021/NN303414U

Reviewer #2 (Remarks to the Author):

The authors developed an integrated tactile/haptic ring with multimodal sensing and feedbacks including the triboelectric and pyroelectric sensor for tactile and temperature perception, and the vibrator and nichrome heater for vibro- and thermo-haptic feedback. With virtual reality technology, they were visually displayed. This work is integrated and interesting. There are several comments as the following:

1. In the abstract, the author mentioned "with the aid of voltage integration processing, continuous finger motion tracking is first achieved via the triboelectric tactile sensor" However, the

reference 27 has provided such a continuous finger motion tracking mode and the authors further improved it in [Nano-Micro Letters, 13 (2021) 51] validating its effectiveness.

2. In line 76, there's a mistake in writing "thst".

3. The best resolution of finger bending control can achieve?

4. Would the signal be different for different persons? Since the ring size is different for individuals and the signal quality may be affected by inappropriate ring size. And how it affect the classification accuracy?

5. The TENG based sensor has ultralow power consumption. What power consumption of the signal processing unit, data communication module, and machine-learning training and optimization system? The authors should provide the power consumption of each unit by experiments and calculation.

Reviewer #3 (Remarks to the Author):

This paper presents an interesting development of augmented tactile/haptic ring with multimodal sensing and feedback capabilities by integrating the triboelectric and pyroelectric sensor for tactile and temperature perception, and the vibrator and nichrome heater for vibro- and thermo-haptic feedback.

Compared with conventional glove type system, this ring type system shows novelty and advantages. Considering recent intense interest in the VR/AR devices, this study may be recommended for publication if the authors can successfully address the following comments,

1) In this study, all these components could be directly driven by a custom processing circuit with low power consumption for wearable/portable scenarios. With the aid of voltage integration processing, continuous finger motion tracking was achieved via the triboelectric tactile sensor on mobile platforms, which also contributes better performance in gesture/object recognition during artificial intelligence analysis. The sensors usually consume less power than other devices and the authors demonstrated well on the low power consumption wearable sensors. However, for the haptic device (referred as 'actuation systems' in this study), especially thermal sensation usually consumes relatively larger power and the power efficiency will be very low. I wonder how is the overall energy consumption efficiency including all the devices from sensors to haptic devices.

2) Continuing the previous comment, this minimalistic designed ring with multimodal sensing and feedback functions shows the merits of a high level of integration and good portability compared to other state-of-art glove-based solutions. the ATH-Ring uses self-powered sensing units and low driven-power feedback elements to achieve low power consumption. Cause the haptic units

(vibrotactile and thermotactile) consume relatively higher energy and the energy generation from the TENG or piezoelectric material will not be sufficient to run the haptic units, they may need external power source or high capacity energy storage device (like bulky battery). However, the overall power needed to run all the units and the way how the energy is supplied (by external wired power source or independent battery, or hybrid (wireless for sensor but wired for haptics)) is not clearly explained. This needs to be provided in the manuscript.

3) The artificial feeling from the tactile feedback system by vibrotactile will be largely dependent on the frequency range of the vibrator cause the mechanoreceptor (SA, FA) in the skin will respond to the different stimulus frequency (Adv. Mater. 30, 1706299 (2018)). What is the frequency range this ring can generate?

4) For the device structure demonstrated in this study, the ERM vibrator is located at the top of each ring to deliver vibration to the entire finger, while the PVDF sensor is attached to the outer surface of the ring to measure the temperature of the object being touched during grasping. I wonder if the vibrator had better be located close to the skin (such as inside the ring, not outside the ring) to provide more vivid haptic information to the skin cause the vibration may be attenuated when it travels through the ring structure to the skin.

5) For the thermotactile system, NiCr wire basically uses a Joule heating to heat up the contacting skin. Small Joule heater itself can increase the temperature with relatively smaller power and fast. However, when the Joule heater is in contact with the skin, it usually needs considerably higher power and response time because skin has a big specific heat (this mean more energy is needed to increase the temperature by same degree). (a) In the supplementary figure 9a, the relationship between the driven power and the final maintained temperature and the response time of the heater corresponding to different driven voltages are provided. I wonder if the temperature is the temperature measurement of the NiCr wire or skin (or the temperature of the parts of the ring directly contacting with the skin) because temperature will be very different when the NiCr wire is in contact or not. (b) The same question applies to the response time (supplementary figure 9b). Is this response time graph for the case when the it is direct contact with skin or not? (c) Along with the temperature information, heat flux measurement information will be very useful to access the heat feeling. (d) The thermal response time for heating is around several or several tens of seconds as shown in supplementary figure 9a,b. However, they show much longer time for cooling back to the original temperature. This may mean that the user still feel the thermal feeling for long time even after the user took off hands from the heated object in the VR system, which will degrade the immersive feeling. Compared with the recent thermoelectric bifunctional heating/cooling based thermotactile device (ref. 53), the response time is much longer. Those actual issues need to be further discussed in the manuscript.

6) The direct control demonstration of the robotic finger with the wirelessly connected sensors is impressive. By integrating five TENG tactile sensors with 3D-printed soft connectors and using a custom IoT module for signal collection and transmission, an advanced manipulator is successfully achieved to implement the real-time and continuous robotic collaborative operation such as the continuous bending control of a robotic finger. Especially, the machine learning enabled gesture/sign language recognition system looks pretty interesting. As the related study in the machine learned sensor, a glove sensor array to identify individual object (Nature 569, 698 (2019)), a single machine

learned sensor could recognize the whole finger motion by machine learning (Nat. Comm., 11, 2149 (2020)) and more. They need to be discussed in the manuscript to provide readers with the recent progresses.

7) Continuing the previous question on the thermohaptic, heating is not the only thermal feeling that the human skin can feel. Beside the hot feeling, cold feeling will be needed for real thermal experience. This issue also needs to be discussed in the manuscript.

8) The authors presented a good introduction on the recent progress in the VR devices including sensors. However, the recent development in the haptics is more extensive than the short discussion in this paper. The authors are suggested to check the reviews on the recent VR/AR progress (iScience 23, 101397 (2020); Adv. Funct. Mater. 31, 2106546 (2021)). There are many other types of haptic devices than those discussed in this study. This may provide a better perspective to the readers.

9) For the actual VR application demonstration, the authors only add one thermo-haptic feedback component to the thumb ring. Is there any specific reason why the thumb is selected for the thermohaptic (instead of other fingers)? Does it provide a better thermal sensation?

Dear Reviewers:

Thank you for your letter and for the reviewers' comments concerning our manuscript entitled "Augmented Tactile/Haptic Rings as Human-Machine Interfaces (HMIs) Aiming for Immersive Interactions". Your comments are all valuable and very helpful for revising and improving our paper, as well as the important guiding significance to our research. We have studied comments carefully and have made the correction which we hope to meet with approval. The main corrections in the paper and the responses to the reviewer's comments are as following:

Reviewer #1 (Remarks to the Author):

The authors propose a multimodal sensing and feedback ring for metaverse applications, which contains TENG- and PyENG-based self-powered sensors as tactile and temperature perception units, and vibrators and nichrome heaters for vibro- and thermo-haptic feedback. All these functionalities could be achieved on minimalistic designed rings and directly driven on wearable/portable platforms, showing a high level of integration compared to other glove-based perception/feedback related solutions. In contrast to the recent research trends in self-powered sensor enabled finger bending monitoring or gesture recognition system (Nat. Electron. 3, 571–578 (2020); Nat. Commun. 12, 5378 (2021)), the newly proposed voltage-integration approach not only makes the TENG sensors capable of monitoring continuous finger motion on mobile platforms, but also show the advantages in continuous sign language interpretation. Besides, a novel interactive virtual chat platform is also successfully demonstrated, to give users an immersive social experience for future metaverse based applications. I think this work do very comprehensive investigations on multimodal perception and feedback system development and the system integration for advanced human-machine interactions. Hence, I recommend the minor revision with the following comments:

Thanks a lot for reviewer's comment. The following is our point-to-point reply along with reviewer's comments.

Comment 1: In Fig. 2, the authors propose to use voltage integral signal from TENG sensors to measure the finger bending angles. Compared to other TENG-based related works, may the authors comment the detailed advantages of this kind of sensing method in a small table to help readers better understand the scientific value of this approach? Can the author comment on what scenarios are suitable for using voltage integral signal or original voltage signal?

Response:

We thank a lot for reviewer's valuable comment. Based on reviewer's suggestion, we draw a small table that compare the state-of-art TENG-based strain sensors for finger motion tracking in terms of sensing mechanism, quantization method, resolution, measurement tool/platform and gesture recognition application as shown in **Table R1**. Based on the information of previous works in **Table R1**, if we want to achieve the continuous tracking based on TENG sensors, we either use a measuring instrument with extremely large internal resistance (ref 1, 2, 10 in **Table R1**), i.e., electrometer, to obtain an approximate open-circuit measurement environment to detect the open-circuit voltage or transferred charge quantity, or use grating-sliding structural sensor (ref 6, 15 in **Table**

R1) to measure the deformation/displacement based on the number of generated peaks. For methods based on the open-circuit measurement environment, the instruments are commonly bulky and expensive, which are not suitable for daily usage and wearable/portable application scenarios. While for the detection method based on the grating-sliding mode sensors, though the signal could be collected by the commercial or customized ADC on portable measurement platforms, the resolution is limited by the size and spacing of the grating electrodes, and is difficult to reach a very high level without advanced fabrication processes, e.g., MEMS process, screen printing, etc., which means high fabrication cost. Besides, even if the size/spacing of the gating can be further reduced, the distinguishability of the signal will be a concern. Essentially, the measurement method based on the grating-sliding mode is not completely continuous because it requires electrodes to be arranged intermittently with certain gaps, where the size/spacing of the grading electrodes determines how much information is lost. As listed in **Table R1**, the current best resolution could be achieved by grating-sliding mode TENG sensor for finger motion tracking is 3.8° . However, in this work, the continuous changes of the muscles during finger bending can be well reflected in the deformation of the pyramid structure to generate the corresponding continuous output as depicted in **Fig. 2** and **Supplementary Fig. 5**, making the resolution can be as low as 1° .

In addition, the proposed voltage integration method can reflect this continuous signal change in a portable platform only by coding or algorithm, without the need for bulky and expensive open-circuit measuring instruments, providing the possibility to realize the real continuous measurement of TENG sensor signal on the mobile terminal, which have not be achieved by previous works (**Table R1**).

Another advantage of using voltage integration method is the interpretation capability of continuous gestures. Current TENG-based gesture recognition works all use the load voltage as the input signal for gesture recognition as listed in **Table R1**, where the action of making a specific gesture will be influenced by the gesture of the previous moment, resulting in the difference in gesture signals between the individual gesture, and the same gesture in a continuous sentence. This phenomenon has been explained clearly in the main text of the article and **Fig. 3f**. Though some works (**ref 3, 7, 16 in Table R1**) have successfully realized the continuous gesture recognition by analyzing the load voltage signal of a whole sentence, this method is not suitable for practical applications considering the various combinations of sentences in sign language and the time and labor costs required to build such a database. However, for our proposed voltage integration method, due to that there is nearly no difference in voltage integration signals between the individual gesture and the corresponding gesture in continuous sentence, we can just build a data set based on the individual gesture, which can help us greatly save the cost of collecting gestures, and improve the universality of the data set for continuous sign language interpretation.

In summary, the signal processing approach based on the voltage integral signal for TENG-based sensory systems can replace the previous methods based on the load voltage amplitude or grating-sliding peak numbers, for continuous stimuli/motion monitoring on portable platforms, thanks to the simple and lightweight measurement environment, i.e., customized ADC, and better sensor performance. Between, it also shows great advantages in the application scenarios of continuous sign language or gestures.

Actual Change:

The concerned content has already been added and highlighted in page 10, line 223-225 and Supplementary Note 4. Table R1 in reply is also provided as Supplementary Table 3 in the Supplementary Information.

Table R1. Comparison of TENG-based strain sensors for finger motion tracking.

Ref	Sensing Mechanism	Materials	Continuous Tracking	Resolution	Measurement tool/platform	Gesture Recognition
1	Grating-Sliding Mode (Peak Number)	PDMS/Water	Yes	45°	Oscilloscope	NA
2	Contact-Separation Mode (Charge Quantity)	FEP/Al	Yes	NA	Electrometer	NA
3	Contact-Separation Mode with Interlocked Structure (Voltage Amplitude)	P(VDF-TrFE)/PDMS	No	10°	Oscilloscope	Limited Continuous Gesture Recognition
4	Contact-Separation Mode with Yarn-based Structure (Voltage Amplitude)	Conductive yarn/ Silicone rubber	No	NA	Electrometer	Single Gesture
5	Contact-Separation Mode with Ladder-shaped Structure (Voltage Amplitude)	PTFE/Al	No	20°	Electrometer	Single Gesture
6	Grating-Sliding Mode (Peak Number)	FEP/Copper	Yes	3.8°	Commercial ADC	NA
7	Contact-Separation Mode with E-textile Based Sensor (Voltage Amplitude)	Nylon/Polyester	No	20°	Customized ADC	Limited Continuous Gesture Recognition
8	Contact-Separation Mode with Arch-shaped Structure (Voltage Amplitude)	PEDOT:PSS coated textile/Silicone rubber	No	30°	Customized ADC	NA
9	Contact-Separation Mode with Arch-shaped Structure (Voltage Amplitude)	CNT-TPE coated textile/Silicone rubber	No	30°	Customized ADC	Single Gesture
10	Contact-Separation Mode with Nestable Arch-shaped Structure (Open-circuit Voltage)	Silicone rubber/Al	Yes	NA	Electrometer	NA
11	Contact-Separation Mode with Microstructured Layers (Voltage Amplitude)	Nylon/PTFE	No	15°	Customized ADC	Single Gesture
12	Contact-Separation Mode with Yarn-based Structure (Voltage Amplitude)	PDMS/Polyester	No	NA	Customized ADC	Single Gesture
13	Contact-Separation Mode with Dome-shaped Sensor	Silicone rubber	No	30°	Customized ADC	Single Gesture

	(Voltage Amplitude)					
14	Contact-Separation Mode (Voltage Amplitude)	Silicone rubber/PDMS	No	1°	Customized ADC	Single Gesture
15	Grating-Sliding Mode with Magnetic Array (Peak Number)	FEP/Copper	Yes	18°	Commercial ADC	NA
16	Contact-Separation Mode with Textile-based Sensor (Voltage Amplitude)	Silicone rubber/Nitrile	No	NA	Customized ADC	Limited Continuous Gesture Recognition
This work	Contact-Separation Mode with Pyramid-structural Layer (Voltage Integration)	Silicone rubber	Yes	1°	Customized ADC	Enhanced Continuous Gesture Recognition

Comment 2: The machine learning enabled gesture/object recognition function relies on the outputs of the TENG tactile sensors. How is the stability of the sensors and the integrated sensory system? After cycles of utilization, is the accuracy still be maintained? Another table comparing the recognition performance of this work and prior similar research is suggested to further highlight the strengths of this work.

Response:

We thank a lot for reviewer's valuable comment. To verify the stability of our developed sensor, we have done the durability test as shown in **Fig. R1**, where the sensor is repeatedly pressed by the force gauge with the same force. The result shows that, after thousand cycles of utilization, the output of the sensor, i.e., the load voltage amplitude and voltage integration value, did not decay. Because the recognition performance of our system is based on the final integration of the voltage output, as long as this value does not change much, the accuracy of the identification will not be affected. Based on the reviewer's suggestion, we have drawn another table comparing the state-of-art wearable sensory systems for gesture/object recognition (**Table R2**). Though it's difficult to accurately compare the performance between these works due to the differences in sensors, collected gestures/objects, and the size of the datasets, it is clear that our work can achieve high-accuracy gesture and object recognition with a small number of sensor nodes, which shows comparable performance among these state-of-art wearable gesture/object recognition system.

Actual Change:

The relevant discussion has been added and highlighted in **page 12, line 273-278**. **Fig. R1** and **Table R2** are provided in the Supplementary Information as **Supplementary Fig. 9** and **Supplementary Table 4**, respectively.

Fig. R1. Durability test of the TENG tactile sensor after 4 hours of repetitive contact-separation process at 40 N.

Table R2. Comparison of wearable sensory systems for gesture/object recognition.

Ref	Type	Sensing Type		ML-enabled Function		Based on Self-powered Sensing Mechanism
		Tactile	Strain	Object Recognition	Gesture Recognition	
17	Glove	Piezoresistive sensor array (548 sensors)	NA	≈ 84% (26 objects)	89.4% (8 gestures)	No
9	Glove	NA	Triboelectric textile sensor (10 sensors)	NA	95.23% (11 gestures)	Yes
13	Glove	Triboelectric dome-shaped tactile sensor (16 sensors)	NA	96.88% (6 objects)	NA	Yes
18	Glove	Fluidic pressure sensor (6 sensors)	Resistive knit sensor (16 sensors)	99.7% (30 objects)	NA	No
12	Glove	NA	Triboelectric yarn-shaped sensor array (5 sensor)	NA	98.63% (11 gestures)	Yes
19	Sleeve	NA	Resistive sensor array (4 sensors)	NA	88.29% (3 gestures)	No
20	E-skin	NA	Resistive skin-like sensor (1 sensor)	NA	96.2% (8 finger motions)	No
21	E-skin	NA	sEMG electrode	NA	97.12% (13 gestures)	No

			arrays on flexible substrates (16 x 4 array)		92.87% (21 gestures)	
16	Glove	Triboelectric textile sensor (3 sensors)	Triboelectric textile sensor (10 sensors)	NA	91.3% (50 sign language words)	Yes
This work	Ring	NA	Triboelectric ring-shaped sensor (5 sensors)	96.56% (8 daily objects)	99.82% (14 gestures)	Yes

Comment 3: The illustrations of the entire device/system shown in Figure 4a and Supplementary Figure 1 appear to differ in the appearance of the TPU connectors. If all the main functionalities are integrated in rings, and “other components just for connection or assembly purposes”, it's better to make it consistent throughout the whole manuscript.

Response:

We appreciate the reviewer’s suggestion. To keep the system throughout the whole manuscript consistent, we have retaken the **Supplementary Video 2-4** based on the device shown in **Supplementary Figure 1**, and replaced some illustrations and related signal figures in **Fig. 4** based on the newly shot videos.

Actual Change:

Supplementary Video 2-4, and **Fig. 4** have been updated as **Supplementary Video 3-5** and **Fig. 4** in the Supplementary Information and manuscript.

Comment 4: In Fig. 4b, the authors show the vibration intensity variation while grasping a soft virtual object. However, in the main text, the authors claim that “Through this kind of control logic and feedback system, the haptic perception of objects under different hardness could be well mimicked.”, may the authors also provide the real-time vibration intensity of grasping rigid virtual objects and compare it with that of the soft one to make the statement more convinced.

Response:

We appreciate the reviewer’s suggestion on the vibration intensity of grasping rigid virtual objects. To respond to this comment, we have provided additional data as the supplementary information of **Fig. 4b** to show the differences in the control logic of vibration intensity for grasping rigid and soft virtual objects as illustrated in Fig. R2. Here, the bending angle is defined as the angle at which it can continue to squeeze after touching the surface of the object, and the relationship between the real-time vibration intensity and the bending angle is plotted in **Fig. R2d**. When touching or pressing a soft

object, because the soft object is deformable, so the vibration intensity increases with the bending angle to provide an actual feeling of squeezing soft objects. However, for pressing rigid objects which are undeformable, the vibration intensity reaches its maximum at the moment of contact with the surface of the object. Based on this difference in the control logic of vibration intensity, the haptic perception of grasping rigid and soft virtual objects can be well mimicked. In addition, the slope of the curve of vibration intensity versus bending angle can be adjusted according to the stiffness of the object, where a larger slope means greater stiffness. Based on this, the varying feedback of touching virtual objects with different stiffness could also be well simulated.

Actual Change:

This discussion has been added and highlighted in page 14, line 322-331. Fig. R2 is also added as Supplementary Fig. 11 in the Supplementary Information.

Fig. R2. Control logic of vibration intensity for grasping rigid and soft virtual objects. The illustrations of pressing **a**, soft and **b**, rigid virtual objects. **c-d**, The relationship between squeezing/pressing angle and vibration intensity for virtual objects with different stiffness.

Comment 5: For the actual application as interactive tools in combination of haptic feedback and sensation, feedback and perception often happen at the same time, and some mutual interferences might be unavoidable, e.g., the vibration of the vibrators will affect the output of the tactile sensor, the temperature change brought by the heater will change the working environment of the tactile sensor. So in this

device, do these interferences exist? If yes, how to avoid such effects and make them work simultaneously?

Response:

Thank you for the reviewer's kind advice. To test whether the interferences between the sensor signal and feedback functions exist, we have done the additional experiments and the results are shown in Fig. R3 and Fig. R4. In Fig. R3, we compare the outputs of the TENG tactile sensor collected by our customized IoT module when the vibrator was not actuated (**Fig. R3a**) and at maximum vibration intensity (**Fig. R3b**). It is clear that though the noise density increases a little bit under the maximum vibration intensity, the maximum amplitude of the noise doesn't change. So the effect of noise can be removed with a threshold of the same size without affecting the output of the sensor considering the huge difference in amplitude between the noise (maximum amplitude: 0.005 V) and valid signal. The interference from the vibration could be ignored in our system mainly because we have added an RC filter circuit (low pass filter: < 10 Hz) in the signal processing circuit (see **Supplementary Fig. 3**), where the hundred-hertz noise from vibrations can be easily filtered out.

Fig. R4 shows the influence of the temperature change on the TENG tactile sensor output. In our design, although the tactile sensor and heater are integrated into one unit, the two parts are separate and spaced apart internally as illustrated in **Fig. R4a**. When the heater is heated to more than 70 °C, though the sensing area will be affected by the temperature due to heat diffusion, the maximum temperature of the sensing part is about 40 °C as shown in **Fig. R4b**. As mentioned in some literature data (Nano Energy, 2014, 4, 453–460; Appl. Phys. Lett., 2015, 106, 013224), the triboelectric output will experience the significant fluctuation after reach above 320K or below 200K. In **Fig. R4c**, we test the TENG tactile sensor outputs when the temperature of the working environment increases from 25 °C to 45 °C, and the result shows that the TENG output will not experience significant fluctuation under such working conditions. Therefore, in our daily use, the temperature of the required thermal feedback will not be too high. In this case, the temperature change generated by the heater will not have too much influence on the output of the sensor.

Actual Change:

The relevant content has been added in page 16, line 375-378, and explained in detail in Supplementary Note 7 in the Supplementary Information. Fig. R3 and Fig. R4 are provided as Supplementary Fig. 15 and Supplementary Fig. 16, respectively.

Fig. R3. The outputs of the TENG tactile sensor collected by our customized IoT module when the vibrator **a**, is not actuated and **b**, at maximum vibration intensity.

Fig. R4. The influence of the temperature change on the TENG tactile sensor output. **a**, Illustration of the TENG tactile sensor with the NiCr metal wire heater. **b**, The maximum temperature of the sensing area when the heater is heated to more than 70 °C. **c**, The outputs of the TENG tactile sensor under the consistent pressure of 40 N when the

temperature of the sensor increases from 25 °C to 45 °C.

Comment 6: What is the response time of the metaverse-based interactive system shown in Fig. 5 (including the sensor/actuator response time and wireless transmission time), i.e., the time from when the command is sent from one user side to when the haptic actuator starts to work from another user side.

Response:

We appreciate the reviewer's comment on response time, and we agree that a fast response is very important for real-time human-machine interfaces to give users a good experience. In our metaverse-based interactive system depicted in **Fig. 5a**, the response process can be divided into two parts. One is the object recognition part, where the object in the real space of a user could be projected into the virtual space on the cloud through the collected sensor signals and machine learning. Another is the real-time control and haptic feedback part, where the user can control the motion of the virtual hand in the metaverse space and feel the virtual object via the vibro- and thermo-haptic feedback functions. The process of the object recognition part is shown in detail in **Fig. R5a**, mainly including the acquisition of tactile and temperature signals when grasping objects, the wireless transmission in the local area network, the real-time object recognition based on machine learning, and the communications with the server twice. Based on the response time of the TENG sensor, PVDF sensor, actuators, wireless transmission, ML recognition and cloud server shown in **Fig. R5c-d** and **Table R3**, we know that the response time of the signal acquisition process is determined by the PVDF sensor due to its longer response time compared to that of the TENG tactile sensor. The total response time of the object recognition part can be calculated as ~ 176 ms. While for the real-time control and haptic feedback part shown in **Fig. R5b**, the whole process contains the acquisition of tactile signals, the wireless transmissions in the local area network twice, and the actuation of vibro- and thermo-haptic feedback units, where the response time of the feedback actuation process is mainly determined by the heater. Because the response time of the heater is determined by the driving voltage and the maximum temperature to be reached. Therefore, in the case of the highest driving power of the platform, and considering the temperature (< 60 °C) to be reached in practical applications, the ideal response time of the real-time control and haptic feedback part could be calculated as ~ 9.13 s. If we combine these two parts into an interactive system, the time for a complete interaction is ~ 9.3s, which is acceptable considering the generally long response time of the thermal feedback function.

Actual Change:

This discussion has been mentioned and highlighted in **page 20, line 457-459**, and explained in detail in **Supplementary Note 9. Fig. R5** and **Table R3** have also been added in the Supplementary Information as **Supplementary Fig. 20** and **Supplementary Table 7**, respectively.

Fig. R5. The response time of the multimodal sensing and feedback platform for metaverse.

The response process of the **a**, object recognition part and **b**, real-time control and haptic feedback part. The response time of the **c**, TENG tactile sensor, **d**, PVDF temperature sensor and **e**, ERM vibrator respectively.

Table R3. The response time of different components in the system.

	TENG Sensor	PVDF Sensor	Vibrator	Heater	Wireless Transmission	ML Recognition	Cloud Server
Response Time	123 ms @ Fast bending 90°	160 ms @ T = 70°C	68 ms	9 s @ ~ 60 °C (Skin)	4.8 ms	~ 1 ms	5 ms

Comment 7: Since the sensor is directly contacted with skin for triboelectrification. Will the pyramid structure cause uncomfortable feeling to user? How about the signal stability for different user with different finger size?

Response:

We thank you for the reviewer’s comments on the device’s comfortability and stability, and these are the key points that should be paid attention to in practical applications. Elastomer materials, e.g., Polydimethylsiloxane (PDMS) and Ecoflex, have been widely utilized for developing wearable TENG devices, e.g., Nano Energy, 2020, 69, 104417; ACS Nano 2018, 12, 11561–11571; Advanced Materials, 2016, 28(45), 10024-10032, etc., due to their high flexibility and stretchability. Between, silicone rubber based on

Ecoflex material commonly shows a lower Young's module compared to that of PDMS, enabling devices with greater comfort in application scenarios where sensors need to be in direct contact with human skin. Since the material itself is quite soft, as long as the rings are properly tight, the deformable pyramid structure will not produce a noticeable foreign body sensation on the skin. Moreover, the pyramid structure has also been proven to help enlarge the sensing range of the tactile sensor, e.g., npj Flexible Electronics, vol. 4, 29, 2020., which is first proposed to be used on the ring and combined with the integral signal processing method to realize the capture of continuous finger movements.

For the issue of the finger size, we agree with the reviewer's opinion that different users have different finger sizes, which will result in the variation of the sensor signal if they use the same size device. As depicted in **Supplementary Fig. 4b-c**, the mismatch of finger and ring size will decrease the finger output, and optimized ring size and pyramid height are needed for different finger sizes or users to provide not only higher sensor output, but also better wearing comfort. However, for actual applications, it's impractical to provide a custom ring size for every finger of every user. A more practical approach is that we provide several sizes from small to large to meet the needs of different users as much as possible, in which case the signal differences are inevitable. The best solution to this issue is to use a software algorithm to calibrate the sensor signal based on each user's finger size on the first use. As shown in **Fig. R6**, different finger sizes will result in the variation of sensor sensitivity with the same size of ring and pyramid structure. However, due to the approximately linear relationship between the bending angle and the output, if we regard the green line in the figure as the standard output of the sensor, then we can easily convert the output of different fingers by a factor to convert to the standard output during calibration. This can be counted as a simple normalization process, which is also a common way to avoid individual errors when doing machine learning, e.g., Nature, 2019, 569(7758): 698-702. As long as all data, both collected in real time and in the database, are normalized, the errors between fingers and individuals can be avoided and will not influence the accuracy of identification. This calibration process only needs to be performed the first time the user uses it and does not affect the ease of use of the device. This method avoids the complexity of calibration at the hardware level and greatly reduces the cost of targeted design.

Actual Change:

The concerned content has been added and highlighted in page 11, line 250-253, and discussed in detail in Supplementary Note 5. Fig. R6 has also been added as Supplementary Fig. 7 in the Supplementary Information. Sensor output regarding the ML in Fig. 3 has been updated with this normalization process to better show the difference between gestures.

Fig. R6. Signal normalization process of the TENG tactile sensor. Illustrations of the TENG tactile sensor when the diameters of the finger are **a**, 15 mm and **b**, 13 mm, respectively, at fixed ring size (22 mm) and pyramid height (3 mm). **c**, The corresponding voltage integration outputs in terms of different bending angles and the normalized signal output to avoid individual differences.

Comment 8: They need to add the description about the "It is the first time that Yang et al. have utilized one tactile sensor to realize the pressure, temperature and material identification. [Wang et al., Sci. Adv. 2020; 6 : eabb9083]" in the first part of the manuscript.

Response:

We thank a lot for reviewer's valuable comment. The suggested paper has been added into the introduction part of the manuscript with the relevant updated discussion to have a broader view to the current research advancements.

Actual Change:

The relevant change can be found in page 4, line 81-84, and this paper is cited ref 33 in the manuscript.

Comment 9: The pyroelectric nanogenerator and sensor's related papers should be cited and described :

- [1] DOI: [10.1021/NL3003039](https://doi.org/10.1021/NL3003039)
- [2] DOI: [10.1002/ADMA.201201414](https://doi.org/10.1002/ADMA.201201414)
- [3] DOI: [10.1021/NL303755M](https://doi.org/10.1021/NL303755M)
- [4] DOI: [10.1021/NN303414U](https://doi.org/10.1021/NN303414U)

Response:

We thank a lot for reviewer's valuable comment. The suggested papers have been cited and described in the introduction to provide a better perspective to the readers.

Actual Change:

These papers have been cited and described as ref 28-30, 32 in page 4, line 78-81.

Reviewer #2 (Remarks to the Author):

The authors developed an integrated tactile/haptic ring with multimodal sensing and feedbacks including the triboelectric and pyroelectric sensor for tactile and temperature perception, and the vibrator and nichrome heater for vibro- and thermo-haptic feedback. With virtual reality technology, they were visually displayed. This work is integrated and interesting. There are several comments as the following:

Thanks a lot for reviewer's comment. The following is our point-to-point reply along with reviewer's comments.

Comment 1: In the abstract, the author mentioned “with the aid of voltage integration processing, continuous finger motion tracking is first achieved via the triboelectric tactile sensor” However, the reference 27 has provided such a continuous finger motion tracking mode and the authors further improved it in [Nano-Micro Letters, 13 (2021) 51] validating its effectiveness.

Response:

To response to this concern, we draw a small table that compare the state-of-art TENG-based strain sensors for finger motion tracking in terms of sensing mechanism, quantization method, resolution, measurement tool/platform and gesture recognition application as shown in **Table R1**. Based on the information of previous works in **Table R1**, if we want to achieve the continuous tracking based on TENG sensors, we either use a measuring instrument with extremely large internal resistance (ref 1, 2, 10 in **Table R1**), i.e., electrometer, to obtain an approximate open-circuit measurement environment to detect the open-circuit voltage or transferred charge quantity, or use grating-sliding structural sensor (ref 6, 15 in **Table R1**) to measure the deformation/displacement based on the number of generated peaks. For methods based on the open-circuit measurement environment, the instruments are commonly bulky and expensive, which are not suitable for daily usage and wearable/portable application scenarios. While for the detection method based on the grating-sliding mode sensors, though the signal could be collected by the commercial or customized ADC on portable measurement platforms, the resolution is limited by the size and spacing of the grating electrodes, and is difficult to reach a very high level without advanced fabrication processes, e.g., MEMS process, screen printing, etc., which means high fabrication cost. Besides, even if the size/spacing of the gating can be further reduced, the distinguishability of the signal will be a concern. Essentially, the measurement method based on the grating-sliding mode is not completely continuous because it requires electrodes to be arranged intermittently with certain gaps, where the size/spacing of the grading electrodes determines how much information is lost. As listed in **Table R1**, the current best resolution could be achieved by grating-sliding mode TENG sensor for finger motion tracking is 3.8° . However, in this work, the continuous changes of the muscles during finger bending can be well reflected in the deformation of the pyramid structure to generate the corresponding continuous output as depicted in **Fig. 2** and **Supplementary Fig. 5**, making the resolution can be as low as 1° . In addition, the proposed voltage integration method can reflect this continuous signal change in a portable platform only by coding or algorithm, without the need for bulky

and expensive open-circuit measuring instruments, providing the possibility to realize the real continuous measurement of TENG sensor signal on the mobile terminal, which have not been achieved by previous works (**Table R1**).

Another advantage of using voltage integration method is the interpretation capability of continuous gestures. Current TENG-based gesture recognition works all use the load voltage as the input signal for gesture recognition as listed in **Table R1**, where the action of making a specific gesture will be influenced by the gesture of the previous moment, resulting in the difference in gesture signals between the individual gesture, and the same gesture in a continuous sentence. This phenomenon has been explained clearly in the main text of the article and **Fig. 3f**. Though some works (ref 3, 7, 16 in **Table R1**) have successfully realized the continuous gesture recognition by analyzing the load voltage signal of a whole sentence, this method is not suitable for practical applications considering the various combinations of sentences in sign language and the time and labor costs required to build such a database. However, for our proposed voltage integration method, due to that there is nearly no difference in voltage integration signals between the individual gesture and the corresponding gesture in continuous sentence, we can just build a data set based on the individual gesture, which can help us greatly save the cost of collecting gestures, and improve the universality of the data set for continuous sign language interpretation.

In summary, the signal processing approach based on the voltage integral signal for TENG-based sensory systems can replace the previous methods based on the load voltage amplitude or grating-sliding peak numbers, for continuous stimuli/motion monitoring on portable platforms, thanks to the simple and lightweight measurement environment, i.e., customized ADC, and better sensor performance. We agree with the reviewer that previous works, i.e., ref 27, already realized the continuous sensing based on TENG sensors, and our work can be seen as an improvement. To avoid controversy, we also adjusted the corresponding sentence in the abstract.

Actual Change:

The concerned content has already been added and highlighted in page 10, line 223-225 and Supplementary Note 4. Table R1 in reply is also provided as Supplementary Table 3 in the Supplementary Information. The corresponding sentence in the abstract has also been modified and highlighted in page 2, line 33-34.

Table R1. Comparison of TENG-based strain sensors for finger motion tracking.

Ref	Sensing Mechanism	Materials	Continuous Tracking	Resolution	Measurement tool/platform	Gesture Recognition
1	Grating-Sliding Mode (Peak Number)	PDMS/Water	Yes	45°	Oscilloscope	NA
2	Contact-Separation Mode (Charge Quantity)	FEP/Al	Yes	NA	Electrometer	NA
3	Contact-Separation Mode with Interlocked Structure (Voltage Amplitude)	P(VDF-TrFE)/PDMS	No	10°	Oscilloscope	Limited Continuous Gesture Recognition
4	Contact-Separation Mode with Yarn-based	Conductive yarn/ Silicone	No	NA	Electrometer	Single Gesture

	Structure (Voltage Amplitude)	rubber				
5	Contact-Separation Mode with Ladder-shaped Structure (Voltage Amplitude)	PTFE/Al	No	20°	Electrometer	Single Gesture
6	Grating-Sliding Mode (Peak Number)	FEP/Copper	Yes	3.8°	Commercial ADC	NA
7	Contact-Separation Mode with E-textile Based Sensor (Voltage Amplitude)	Nylon/Polyester	No	20°	Customized ADC	Limited Continuous Gesture Recognition
8	Contact-Separation Mode with Arch-shaped Structure (Voltage Amplitude)	PEDOT:PSS coated textile/Silicone rubber	No	30°	Customized ADC	NA
9	Contact-Separation Mode with Arch-shaped Structure (Voltage Amplitude)	CNT-TPE coated textile/Silicone rubber	No	30°	Customized ADC	Single Gesture
10	Contact-Separation Mode with Nestable Arch-shaped Structure (Open-circuit Voltage)	Silicone rubber/Al	Yes	NA	Electrometer	NA
11	Contact-Separation Mode with Microstructured Layers (Voltage Amplitude)	Nylon/PTFE	No	15°	Customized ADC	Single Gesture
12	Contact-Separation Mode with Yarn-based Structure (Voltage Amplitude)	PDMS/Polyester	No	NA	Customized ADC	Single Gesture
13	Contact-Separation Mode with Dome-shaped Sensor (Voltage Amplitude)	Silicone rubber	No	30°	Customized ADC	Single Gesture
14	Contact-Separation Mode (Voltage Amplitude)	Silicone rubber/PDMS	No	1°	Customized ADC	Single Gesture
15	Grating-Sliding Mode with Magnetic Array (Peak Number)	FEP/Copper	Yes	18°	Commercial ADC	NA
16	Contact-Separation Mode with Textile-based Sensor (Voltage Amplitude)	Silicone rubber/Nitrile	No	NA	Customized ADC	Limited Continuous Gesture Recognition
This work	Contact-Separation Mode with Pyramid-structural Layer (Voltage Integration)	Silicone rubber	Yes	1°	Customized ADC	Enhanced Continuous Gesture Recognition

Comment 2: In line 76, there's a mistake in writing "thst".

Response:

We apologize for the mistake in our manuscript. Thanks for reviewer's advice and we have carefully gone through the whole text, and modified all the typo errors.

Actual Change:

The typo and grammar errors have been modified and highlighted in red throughout the manuscript.

Comment 3: The best resolution of finger bending control can achieve?**Response:**

We thank a lot for reviewer's valuable comment. To help readers better understand the performance of our developed tactile sensor, we have done additional test and added the result into **Supplementary Fig. 5 (Fig. R7)**. As depicted in **Fig. R7b**, when the bending angle increases from 20° to 30°, even measured at intervals of 1°, the variation of the bending degree can still be clearly distinguished, showing the strong perceiving ability of our developed TENG tactile sensor. Compared with other similar work listed in the **Table R1**, the resolution of one degree is also the highest level that can be achieved so far, which is more than enough for daily use. The sensing performance of the TNEG tactile sensor is also shown in **Supplementary Video 1**.

Actual Change:

The relevant discussion has been added and highlighted in page 10, line 220-223. **Supplementary Fig. 5** has also been updated with the new information shown in **Fig. R7**. **Supplementary Video 1** has been added in the Supplementary Information.

Fig. R7. **a**, The proportional relationship between the transfer charge and the voltage integration value in terms of different finger bending angles. **b**, The outputs of the TENG tactile sensor when the bending angle is increased from 20° to 30° and measured at 1° intervals.

Comment 4: Would the signal be different for different persons? Since the ring size is different for individuals and the signal quality may be affected by inappropriate ring size. And how it affect the classification accuracy?

Response:

We thank a lot for reviewer’s valuable comment. For the issue of the finger size, we agree with the reviewer’s opinion that different users have different finger sizes, which will result in the variation of the sensor signal if they use the same size device. As depicted in **Supplementary Fig. 4b-c**, the mismatch of finger and ring size will decrease the finger output, and optimized ring size and pyramid height are needed for different finger sizes or users to provide not only higher sensor output, but also better wearing comfort. However, for actual applications, it’s impractical to provide a custom ring size for every finger of every user. A more practical approach is that we provide several sizes from small to large to meet the needs of different users as much as possible, in which case the signal differences are inevitable. The best solution to this issue is to use a software algorithm to calibrate the sensor signal based on each user’s finger size on the first use. As shown in

Fig. R6, different finger sizes will result in the variation of sensor sensitivity with the same size of ring and pyramid structure. However, due to the approximately linear relationship between the bending angle and the output, if we regard the green line in the figure as the standard output of the sensor, then we can easily convert the output of different fingers by a factor to convert to the standard output during calibration. This can be counted as a simple normalization process, which is also a common way to avoid individual errors when doing machine learning, e.g., Nature, 2019, 569(7758): 698-702. As long as all data, both collected in real time and in the database, are normalized, the errors between fingers and individuals can be avoided and will not influence the accuracy of identification. This calibration process only needs to be performed the first time the user uses it and does not affect the ease of use of the device. This method avoids the complexity of calibration at the hardware level and greatly reduces the cost of targeted design.

Actual Change:

The concerned content has been added and highlighted in page 11, line 250-253, and discussed in detail in Supplementary Note 5. Fig. R6 has also been added as Supplementary Fig. 7 in the Supplementary Information.

Fig. R6. Signal normalization process of the TENG tactile sensor. Illustrations of the TENG tactile sensor when the diameters of the finger are **a**, 15 mm and **b**, 13 mm, respectively, at fixed ring size (22 mm) and pyramid height (3 mm). **c**, The corresponding voltage integration outputs in terms of different bending angles and the normalized signal output to avoid individual differences.

Comment 5: The TENG based sensor has ultralow power consumption. What power

consumption of the signal processing unit, data communication module, and machine-learning training and optimization system? The authors should provide the power consumption of each unit by experiments and calculation.

Response:

We thank a lot for reviewer’s valuable comment. We agree that power consumption efficiency is a quite important parameter for smart devices, especially wearable/portable devices, considering the limited lifespan of batteries and the annoying replacement and recharging process. Due to that the sensors used in our system, i.e., the TENG tactile sensor and the PVDF temperature sensor, are all self-powered sensors that can generate the sensor signal by self-generated energies and do not need the external power supply, so the power consumption of the integrated wearable system mainly comes from the signal processing unit, wireless data communication module and the haptic feedback units (vibrators and heaters). As for machine learning, optimization systems and VR systems, these are all carried out through IoT on local terminals or cloud with supercomputing power and external power supply. In this case, the power consumption generated by these functions is less important to our wearable system. The power consumption and the number of each unit have been listed in **Table R4**, where the power consumption of the haptic units accounts for the main part. Considering the maximum power consumption of the actuators in the actual application scenario, the peak overall power consumption can be calculated as 1.95 W, where all vibrators are at maximum vibration intensity and the heater is heated to around 55 °C. As for the application scenarios that do not need haptic feedback functions, the overall power consumption is around 0.25 W, which is quite low for real-time HMIs with high sampling speed thanks to the self-powered sensors utilized in the system.

Actual Change:

The relevant discussion has been added and highlighted in page 20, line 457-458, and explained in detail in Supplementary Note 6. Table R4 is provided as Supplementary Table 6 in the Supplementary Information.

Table R4. Power consumption of each unit in the system.

	Vibrator	Heater	Signal processing unit	Data communication module	Overall
Unit Number	5	1	1	1	
Power consumption	0.18 w @ Maximum vibration intensity	0.81 w @ 55 °C (Skin)	0.09 w	0.15 w	1.95 w (Peak)

Reviewer #3 (Remarks to the Author):

This paper presents an interesting development of augmented tactile/haptic ring with multimodal sensing and feedback capabilities by integrating the triboelectric and pyroelectric sensor for tactile and temperature perception, and the vibrator and nichrome heater for vibro- and thermo-haptic feedback.

Compared with conventional glove type system, this ring type system shows novelty and advantages. Considering recent intense interest in the VR/AR devices, this study may be recommended for publication if the authors can successfully address the following comments,

Thanks a lot for reviewer's comment. The following is our point-to-point reply along with reviewer's comments.

Comment 1: In this study, all these components could be directly driven by a custom processing circuit with low power consumption for wearable/portable scenarios. With the aid of voltage integration processing, continuous finger motion tracking was achieved via the triboelectric tactile sensor on mobile platforms, which also contributes better performance in gesture/object recognition during artificial intelligence analysis. The sensors usually consume less power than other devices and the authors demonstrated well on the low power consumption wearable sensors. However, for the haptic device (referred as 'actuation systems' in this study), especially thermal sensation usually consumes relatively larger power and the power efficiency will be very low. I wonder how is the overall energy consumption efficiency including all the devices from sensors to haptic devices.

Response:

We thank a lot for reviewer's valuable comment. We agree that power consumption efficiency is a quite important parameter for smart devices, especially wearable/portable devices, considering the limited lifespan of batteries and the annoying replacement and recharging process. Because our system is mainly composed of sensing and feedback units, so this problem is mainly discussed from three aspects: sensor, vibro-haptic feedback unit and thermo-haptic feedback unit:

- 1) In our system, the TENG tactile sensor and PVDF temperature sensor are all based on the nanogenerator that can convert the energy from the human body or in the ambient into the signal, and do not need the power supply. Compared to current commercial wearable sensors whose power consumption is in the range of 3-300 μ W (ref: iScience, 2020, 23(8): 101360), the self-powered sensors in our system are undoubtedly more energy efficient.
- 2) For vibro-haptic feedback unit, based on the comparison of different vibrators in the recent related work (Nat. Electron. (2022), <https://doi.org/10.1038/s41928-022-00765-3>), the ERM vibrator can induce larger vibration intensity under the same operation power compared to other types of vibrators, i.e., linear resonance actuator (LRA), piezoelectric actuator and voice coil actuator (Nature, 2019, 575(7783): 473-479). This can help to create larger and more robust feedback sensation more efficiently. Between, for other vibrators that need to vibrate at a

fixed resonant frequency, the ERM vibrator changes both the vibration frequency and amplitude under different supply voltages, which introduces a stronger change of feeling for users (2016 6th IEEE International Conference on Biomedical Robotics and Biomechatronics (BioRob). IEEE, 2016: 1266-1271).

- 3) For thermo-haptic feedback devices, we have drawn a table to compare the state-of-art works for wearable scenarios (**Table R5**), including the heating technologies based on the Joule heating and thermoelectric. As listed in **Table R5**, our solution based on the NiCr wire can provide a somatosensory temperature close to 55 °C under the power supply voltage of 1.8 V, which shows good energy efficiency compared to other related works.

Between, we have also tested the power consumption of each unit in our system, including the vibrator, heater, signal processing unit and data communication module (**Table R4**). Because our sensors are self-powered, so they can be counted as zero-power components. It is clear that the power consumption of the thermo-haptic unit accounts for the main part of the overall system, which means the lowest energy consumption efficiency compared to other functional units in the system.

When compared with other similar HMIs with multifunctional sensing and haptic-feedback capabilities, because the overall functions of the integrated system are not completely consistent, we cannot use a unified standard to measure and compare the overall energy efficiency. However, as stated in the first three paragraphs, the sensing and feedback units in our system have good energy efficiency when taken out individually and compared with other works, so the overall energy efficiency of the system should also be at a good level.

Actual Change:

The relevant discussion has been added and highlighted in page 20, line 457-459, and explained in detail in Supplementary Note 6. Table R4 and Table R5 are provided as Supplementary Table 6 and Supplementary Table 5, respectively, in the Supplementary Information.

Table R5. Comparison of state-of-art thermo-haptic feedback technologies for wearable scenarios.

Ref	Type	Mechanism	Material	Power Supply	Wearability	Heating/Cooling
22	E-skin	Joule Heating	Ag nanowire	2.8 - 6.2 V @ 50 °C	Stretchable & Flexible	Heating
23	Wrist	Joule Heating	Ag nanowire	3.7 V @ 37 °C (Skin)	Stretchable & Flexible	Heating

24	Glove	Joule Heating	Cu nanowire	4 V @ 60 °C	Stretchable & Flexible	Heating
25	Sleeve	Joule Heating	Carbon nanotube fiber	7 V @ 48 °C (Skin)	Flexible	Heating
26	Sleeve	Joule Heating	Carbon nanotube fiber	9 V @ 42 °C (Skin)	Flexible	Heating
27	Glove	Joule Heating	PEDOT/PSS fiber	9 V @ 36 °C (Skin)	Flexible	Heating
28	Electronic tattoos	Joule Heating	Semi-liquid- metal Ni- EGaIn	0.4 A @ 45.7 °C (Skin)	Stretchable & Flexible	Heating
29	Kneepad	Joule Heating	Liquid-metal galinstan	2 V @ 50 °C	Stretchable & Flexible	Heating
30	Glove	Joule Heating	Liquid-metal eGaIn	1 W @ 80 °C	Stretchable & Flexible	Heating
31	Vest	Thermoelectric	Bi ₂ Te ₃	180 mW @ Δ 6 °C (Skin)	Flexible (Thickness ≈ 6 mm)	Cooling
32	Patch	Thermoelectric	Bi ₂ Te ₃	5 V 1.33 A @ Δ 5 °C (Skin)	Flexible (Thickness ≈ 10 mm)	Cooling
33	Glove	Thermoelectric	Bi ₂ Te ₃	0.6 A @ ~ 35°C 1.5 A @ ~ 12 °C	Stectchable (Thickness ≈ 2 mm)	Heating & Cooling
This work	Ring	Joule Heating	NiCr wire	0.45 A 1.8 V @ 55 °C (Skin)	Flexible	Heating

Table. R4. Power consumption of each unit in the system.

	Vibrator	Heater	Signal processing unit	Data communication module	Overall
Unit Number	5	1	1	1	

Power consumption	0.18 w @ Maximum vibration intensity	0.81 w @ 55 °C (Skin)	0.09 w	0.15 w	1.95 w (Peak)
-------------------	--------------------------------------	-----------------------	--------	--------	---------------

Comment 2: Continuing the previous comment, this minimalistic designed ring with multimodal sensing and feedback functions shows the merits of a high level of integration and good portability compared to other state-of-art glove-based solutions. the ATH-Ring uses self-powered sensing units and low driven-power feedback elements to achieve low power consumption. Cause the haptic units (vibrohaptic and thermohaptic) consume relatively higher energy and the energy generation from the TENG or pyroelectric material will not be sufficient to run the haptic units, they may need external power source or high capacity energy storage device (like bulky battery). However, the overall power needed to run all the units and the way how the energy is supplied (by external wired power source or independent battery, or hybrid (wireless for sensor but wired for haptics)) is not clearly explained. This needs to be provided in the manuscript.

Response:

We thank a lot for reviewer’s valuable comment. As mentioned in the response to comment 1 that the sensors used in our system, i.e., the TENG tactile sensor and the PVDF temperature sensor, are all self-powered sensors that can generate the sensor signal by self-generated energies and do not need the external power supply, so the power consumption of the integrated wearable system mainly comes from the signal processing unit, wireless data communication module and the haptic feedback units (vibrators and heaters). As for machine learning, optimization systems and VR systems, these are all carried out through IoT on local terminals or cloud with supercomputing power and external power supply. In this case, the power consumption generated by these functions is less important to our wearable system. The power consumption and the number of each unit have been listed in Table R4, where the power consumption of the haptic units accounts for the main part. Considering the maximum power consumption of the actuators in the actual application scenario, the peak overall power consumption can be calculated as 1.95 W, where all vibrators are at maximum vibration intensity and the heater is heated to around 55 °C when attached to skin. As for the application scenarios that do not need haptic feedback functions, the overall power consumption is around 0.25 W, which is quite low for real-time HMIs with high sampling speed thanks to the self-powered sensors utilized in the system.

For the power supply, all these functional units could be directly driven by an independent battery (output voltage: 5V) as illustrated in Fig. R7a-c, showing good portability of our developed device for daily wearable application scenarios.

Actual Change:

The content related to the system power consumption has been added and highlighted in page 20, line 457-459, and explained in detail in Supplementary Note 6. Supplementary Fig. 3 has been updated with the newly added information shown in Fig. R8b-c. The discussion about the power supply has been mentioned and highlighted in page 8, line 179-181.

Fig. R8. The Architecture of the entire system. **a**, The detailed circuit schematic shows all connections between the sensing/feedback units and the IoT module. The illustration of the integrated system with the **b**, IoT module (front side) and **c**, power supply battery (back side).

Comment 3: The artificial feeling from the tactile feedback system by vibro haptic will be largely dependent on the frequency range of the vibrator cause the mechanoreceptor (SA, FA) in the skin will respond to the different stimulus frequency (Adv. Mater. 30, 1706299 (2018)). What is the frequency range this ring can generate?

Response:

We thank a lot for reviewer’s valuable comment. As depicted in related research (Nature materials, 2016, 15(9): 937-950), there are four types of mechanoreceptors in human skin, where slow adapting receptors (SA-I and SA-II) respond to static pressures and skin stretch, FA-I receptors response to low-frequency (5-50 Hz) stimuli and help to measure object slip, edges and fine features, and FA-II receptors measure high-frequency vibrations (up to 400 Hz). To get the more accurate vibration data in our application scenario, instead of directly measuring the vibration amplitude of the vibrator itself, we have done the additional test by fixing the vibrator on the TPU ring and achieved a more complete vibration profile via a laser vibrometer (VIB-A-510, Polytec). The result is shown in Fig. R8. In our case, the vibration frequency range of the vibrator is measured as 130-230 Hz, which is within the sensing range of FA-II as vibration stimuli. Many works have demonstrated the feasibility of using vibration stimuli to mimic the touch-related tactile sensation for VR applications, e.g., ref 13, 14 in **Supplementary Table 1**. Actually, the perceptual mechanism behind the experience of holding an object is mainly from kinesthetic components, which need feedback technologies to offer significant static forces to lock the finger position and achieve a more realistic sense of damping for grasping objects. However, current kinesthetic haptic feedback technologies, e.g., pneumatic, hydraulic, electromechanical, etc., usually reveal the drawbacks of bulky

volume and large supply power (ref 9-11 in **Supplementary Table 1**), which are not suitable to be used on portable platforms. Although only using vibration to simulate the perception of grasping or touching an object is not quite intuitive from the perspective of the human biological perception system, the relatively loose working conditions of vibrators make it possible to apply them to portable platforms. Under a certain learning cost, i.e., linking the touch force to the vibration intensity, this approach can also provide a good feedback experience for users. Between, as mentioned in the response to comment 1, compared to other types of vibrators, i.e., linear resonance actuator (LRA), piezoelectric actuator and voice coil actuator, ERM vibrator can induce much larger vibration displacement to create a more robust feedback sensation. Between, other vibrators need to vibrate at a fixed resonant frequency, while the ERM changes both the vibration frequency and amplitude under different supply voltages, which introduces a stronger change of feeling for users.

In our current work, we mainly use the change in vibration intensity to simulate the change in force when touching or squeezing the virtual object. As for providing vibrations of specific frequencies to simulate the most intuitive sense of human biology to achieve more complex feedback functions, e.g., measuring object slip, edges and fine features, etc., limited by the current wearable vibro-haptic feedback technology, i.e., LRA, piezoelectric and voice coil actuator need to vibrate at a fixed resonant frequency and ERM vibrator also cannot provide strong vibration at low frequencies, we didn't adopt such a solution in the current work. But we agree with the reviewer that this is a valuable research direction for haptic feedback in the future, and we will also work towards this aspect in future work.

Actual Change:

The relevant content about the vibration frequency has been added and highlighted in page 14, line 312-316. Supplementary Fig. 10 has also been replaced by Fig. R9 with the newly added frequency information. Fig. 4a(ii) has been revised to show the relationship between the vibration intensity detected by the piezoelectric sensor and the driven voltage.

Fig. R9. The actual a, vibration amplitudes and b, frequencies of the ERM vibrator under different supply voltages.

Comment 4: For the device structure demonstrated in this study, the ERM vibrator is located at the top of each ring to deliver vibration to the entire finger, while the PVDF sensor is attached to the outer surface of the ring to measure the temperature of the object being touched during grasping. I wonder if the vibrator had better be located close to the skin (such as inside the ring, not outside the ring) to provide more vivid haptic information to the skin cause the vibration may be attenuated when it travels through the ring structure to the skin.

Response:

We thank a lot for reviewer's valuable comment. We agree with the reviewer's opinion that the vibrator can deliver more vivid haptic information to the skin if placed close to the skin. From the point of view of human touch sensation or artificial feeling, the vibrator is best placed close to the finger pulp rather than the back of the finger to simulate the tactile sensation of touching/grasping the object. However, in our current device design, the tactile sensor detects the finger bending motion based on the muscle swelling, where the deformation of the muscles is mainly concentrated in the finger pulp area. If we attach a relatively large vibrator to the finger pulp, it will occupy a large sensing area, which will cause a huge influence on the tactile sensor signal. Between, the rigid structure of the vibrator may also be uncomfortable to the skin. Due to the limitations of current vibration feedback technology and considering the integrity of the whole device, we chose to attach the vibrator to the outer surface and make it deliver the vibration to the whole ring. Although in this case the vibration will be attenuated, it is still strong enough to provide a varying vibration intensity and noticeable difference in haptic information for users to perceive. In the future, improvements in vibration technology will be a focus of our research to provide sufficient and vivid feedback in a smaller volume, e.g., MEMS-based haptic feedback units, without compromising other functional components and also ensuring wearing comfort.

Actual Change:

The concerned content has been added and highlighted in page 13, line 303-308.

Comment 5: For the thermohaptic system, NiCr wire basically uses a Joule heating to heat up the contacting skin. Small Joule heater itself can increase the temperature with relatively smaller power and fast. However, when the Joule heater is in contact with the skin, it usually needs considerably higher power and response time because skin has a big specific heat (this mean more energy is needed to increase the temperature by same degree). (a) In the supplementary figure 12a, the relationship between the driven power and the final maintained temperature and the response time of the heater corresponding to different driven voltages are provided. I wonder if the temperature is the temperature measurement of the NiCr wire or skin (or the temperature of the parts of the ring directly contacting with the skin) because temperature will be very different when the NiCr wire is in contact or not. (b) The same question applies to the response time (supplementary figure 12b). Is this response time graph for the case when the it is direct contact with skin or not? (c) Along with the temperature information, heat flux measurement information will be

very useful to access the heat feeling. (d) The thermal response time for heating is around several or several tens of seconds as shown in supplementary figure 12a,b. However, they show much longer time for cooling back to the original temperature. This may mean that the user still feel the thermal feeling for long time even after the user took off hands from the heated object in the VR system, which will degrade the immersive feeling. Compared with the recent thermoelectric bifunctional heating/cooling based thermohaptic device (ref. 53), the response time is much longer. Those actual issues need to be further discussed in the manuscript.

Response:

We thank you for the reviewer's comments on the response time of the NiCr wire heater. We apologize for not clearly specifying the measurement environment of the response time. The results were achieved by putting the integrated unit of tactile sensor and heater (**Supplementary Fig. 1b**) on a TPU substrate, with another side exposed to the air. The temperature profile here is the changing temperature of this unit under different driven voltage, which was collected by using a commercial IR camera and focusing on the top surface. However, we found some errors in the processing and calibration of the previous data. Thanks again for the reviewer's comments, which let us notice the issue in time. We have made corrections and updates in **Supplementary Fig. 12**. Between, for the actual application scenario, the side with the heater is actually in contact with the skin and not exposed to the air, and this may result in a difference in the temperature profile due to that human skin has a big specific heat. To this issue, we have done additional tests on the temperature response of our heater by putting the heater (silicone rubber encapsulated) on a TPU substrate and contacting the heating side with human skin. A small thermistor is placed between the skin and heating surface to measure the temperature profile. The newly achieved results are shown in **Fig. R9**. Instead of measuring the temperature of the heater by the IR camera, here we detect the more accurate temperature of the finger and silicone rubber contact surface by the thermistor, which is closer to the real temperature felt by the fingers. The result depicts that, under the same applied power, the achieved temperature of the contact surface is slightly lower than that of the NiCr wire when the heating unit is exposed to the air. Between, the response time is also much longer, considering the relatively poor thermal conductivity of silicone rubber and the big specific heat of the human body. However, under the maximum power of the IoT platform, this temperature can still be quickly heated up to $\sim 62\text{ }^{\circ}\text{C}$ in 9.4 s, which is sufficient in practical applications that need fast response. Based on the reviewer's suggestion, we have also tested the heat flux information of our heater by using a commercial heat flux sensor with a device sandwiched between a TPU substrate and a heat flux sensor under a finger press. The result plotted in **Fig. R10a** shows that the heat flux increases with the applied power, and the heat can be easily transferred to skin under certain supply power. Between, the approximately linear relationship between the heat flux and the heater temperature (**Fig. R10b**) enables us to control the heat flux simply by controlling the temperature of the heater, which could also be used to mimic various thermal feelings in specific applications, e.g., differentiating objects with different thermal conductivity at the same temperature (Journal of Materials Chemistry A, 2020, 8(17): 8281-8291), etc.

For the issue of long cooling time, we agree with the reviewer that the response time is

much longer when compared with thermoelectric devices (Adv. Funct. Mater. 2020, 30, 1909171) that can actively dissipate heat. As shown in Fig. R9, it takes around 118 s for the interface temperature cools from 55 °C to 35 °C due to the heat accumulated in the silicone rubber, heat of the human body and the poor heat dissipation environment inside the highly integrated device. This is one limitation of our current design, and one possible solution to this issue is to use thermoelectric-based haptic devices with active cooling mode as the reviewer suggested. However, considering the relatively bulky volume of the thermoelectric devices and the necessary heat sink to maintain the cooling efficiency, it takes effort to be able to integrate all the parts well in a tiny ring without affecting the functionality of the other components, e.g., the tactile sensor. In addition, the relatively high power consumption of the cooling mode should also be a concern considering the wearable/portable application scenarios. This can be seen as a further improvement of our current design, and the actual issue is also mentioned in the manuscript.

Actual Change:

The discussion regarding the response time have been updated and highlighted in page 15, line 347-352, and Fig. R10 is provided as Supplementary Fig. 13. Supplementary Fig. 12 has been recalibrated and updated in the Supplementary Information. The content related to the heat flux has been added in page 16, line 362-369. Fig. R11 has also been added as Supplementary Fig. 14 in the Supplementary Information. The detailed measurement set up of the response time and heat flux has been clearly stated in page 23-24, line 544-551 in the Methods section. The issue of the long cooling back time is mentioned in page 16, line 369-371.

Fig. R10. The response time of the thermo-haptic feedback unit when placed on a TPU substrate, with another side in contact with skin. a, The temperature response of the contacting interface corresponding to different driven voltage. b, The rapid heating curve of the contacting interface for fast response scenario, 61.9 °C within 9.4 s with driven power of ~3.5 W.

Fig. R11. The heat flux of the thermo-haptic feedback unit when placed on a TPU substrate, with another side in contact with skin. a, The heat flux of the heating surface under different driven voltage. b, The relationship between the temperature and heat flux of the heating surface.

Comment 6: The direct control demonstration of the robotic finger with the wirelessly connected sensors is impressive. By integrating five TENG tactile sensors with 3D-printed soft connectors and using a custom IoT module for signal collection and transmission, an advanced manipulator is successfully achieved to implement the real-time and continuous robotic collaborative operation such as the continuous bending control of a robotic finger. Especially, the machine learning enabled gesture/sign language recognition system looks pretty interesting. As the related study in the machine learned sensor, a glove sensor array to identify individual object (Nature 569, 698 (2019)), a single machine learned sensor could recognize the whole finger motion by machine learning (Nat. Comm., 11, 2149 (2020)) and more. They need to be discussed in the manuscript to provide readers with the recent progresses.

Response:

We thank a lot for reviewer’s valuable comment. The suggested paper has been added into the introduction part of the manuscript with the relevant updated discussion to have a broader view to the current research advancements.

Actual Change:

The suggested two papers have been cited as ref 48, 49 and discussed in page 5, line 91-94.

Comment 7: Continuing the previous question on the thermohaptic, heating is not the only thermal feeling that the human skin can feel. Beside the hot feeling, cold feeling will be needed for real thermal experience. This issue also needs to be discussed in the manuscript.

Response:

Thank you for the reviewer's kind advice. We agree with the reviewer's opinion that besides the thermal feeling, cold feeling is also very important to provide a more complete thermal experience. Current possible solutions to provide simulated cold feeling for wearable scenarios are mainly based on thermoelectric devices (see ref 31-33 in **Table R5**). Though these works have successfully developed flexible TE devices by embedding small TE pillars with flexible elastomer substrates, there are still some limitations in wearing comfort, considering the non-negligible stiffness and thickness of the TE pillars themselves. Between, to ensure the cooling efficiency of the TE devices, a heat sink is always needed to exchange the heat with the surrounding air at the heated junction (Advanced Functional Materials, 2021, 31(39): 2007376). However, heat sinks are typically bulky and limit them in wearable applications. Though works have shown the possibility of using TE devices to cool the skin surface without the heat sink (ref 31-32 in **Table R5**), the decreased temperature is limited to $\sim 5\text{ }^{\circ}\text{C}$, which is not enough for thermo-haptic applications. The most recent work (ref 33 in **Table R5**) shows a thermal VR/AR glove based on a highly stretchable TE device (Thickness $\approx 2\text{ mm}$) that can achieve 15 and 40 $^{\circ}\text{C}$ under cooling and heating mode respectively with wearable heat sinks (phase-change material (PCM) encapsulated in a Spandex textile). However, the rigid PCM still stiffens the entire TE device and makes the whole device bulkier, greatly reducing the device's wearability.

To our highly integrated ring, the space inside the entire device is very limited. Considering the relatively bulky volume of the TE devices and the necessary heat sink to maintain the cooling efficiency, it takes effort to be able to integrate all the parts well without affecting the functionality of the other components, e.g., the tactile sensor. In addition, the relatively high power consumption of the cooling mode should also be a concern considering the wearable/portable application scenarios. Of course, we agree that cooling capacity is an important function for thermo-haptic devices to enable a more complete thermal experience, especially for VR/AR applications, and have been demonstrated well by recent thermoelectric-based works to provide efficient cold feedback to a relatively large area of the body surface, e.g., palm and chest. However, for our highly integrated ring system, how to integrate the thermoelectric unit and make it work efficiently is still a challenge. We will work in this direction and further improve the function of thermal feedback in our device in the future.

Actual Change:

The limitation of the current design in terms of the cooling mode has been added in page 16, line 369-371.

Comment 8: The authors presented a good introduction on the recent progress in the VR devices including sensors. However, the recent development in the haptics is more extensive than the short discussion in this paper. The authors are suggested to check the reviews on the recent VR/AR progress (iScience 23, 101397 (2020); Adv. Funct. Mater. 31, 2106546 (2021)). There are many other types of haptic devices than those discussed in this study. This may provide a better perspective to the readers.

Response:

We thank a lot for reviewer's valuable suggestions. For the review (iScience 23, 101397 (2020)) about the display technologies, we cite it when mentioning the VR display in the first paragraph of the introduction. We didn't add too much discussion on this part due to that our article is more related to the somatosensory sensation. Between, we have carefully checked the haptic-feedback related content in the review (Adv. Funct. Mater. 31, 2106546 (2021)), and added more discussion about the current haptic technologies when introducing the haptic. More related papers have also been cited to provide a better perspective to the readers.

Actual Change:

Review (iScience 23, 101397 (2020)) has been cited as ref 4 in page 3, line 49. More discussion and reference papers about the haptic-feedback technologies have been added and highlighted in page 5, line 96-113.

Comment 9: For the actual VR application demonstration, the authors only add one thermo-haptic feedback component to the thumb ring. Is there any specific reason why the thumb is selected for the thermohaptic (instead of other fingers)? Does it provide a better thermal sensation?

Response:

Thank you for the reviewer's kind advice. The reason we chose to add the thermo-haptic feedback unit to the thumb ring is that in our specific application scenario of grasping objects, the thumb is the finger that is always used during grasping motion. Of course, if each finger is equipped with a feedback unit, the best feedback experience can be achieved. This is just an effort of engineering and can be easily realized. However, considering the large power consumption of the thermo-haptic feedback unit compared to other components as shown in **Table R4**, we only used one thermo-haptic feedback unit for achieving a low-power and long-sustainable IoT system for wearable/portable application scenarios, where the thermal sensation on the thumb is sufficient for all grasping tasks and most daily usage scenarios. For tasks that need the thermo-haptic feedback to individual fingers, we can add more units but at the cost of very high power consumption, which is one of the main limitations of our current thermo-haptic feedback technology. In the future, we will spend more effort on the development of low-power thermal feedback components, so that on the premise of realizing a more complete thermal feedback function, the power consumption of the entire system can also be maintained low.

Actual Change:

The relevant discussion has been added and highlighted in page 15-16, line 352-355 and line 371-373.

REVIEWERS' COMMENTS

Reviewer #1 (Remarks to the Author):

The revisions are satisfactory. I strongly recommend to accept this manuscript at current revised version.

Reviewer #2 (Remarks to the Author):

It has been revised according to the comments and improved. I recommend accepting it as it is.

Reviewer #3 (Remarks to the Author):

The authors responded well to the comments. This should be ready for publication.

Dear Reviewers:

Thank you for your letter and for the reviewers' comments concerning our manuscript entitled "Augmented Tactile/Haptic Rings as Human-Machine Interfaces (HMIs) Aiming for Immersive Interactions". Your comments are all valuable and very helpful for revising and improving our paper, as well as the important guiding significance to our research.

Reviewer #1 (Remarks to the Author):

The revisions are satisfactory. I strongly recommend to accept this manuscript at current revised version.

Thank you very much for your positive comments on the manuscript.

Reviewer #2 (Remarks to the Author):

It has been revised according to the comments and improved. I recommend accepting it as it is.

We thank a lot for reviewer's acceptance to our manuscript. We will keep spending enough effort to make more contributions.

Reviewer #3 (Remarks to the Author):

The authors responded well to the comments. This should be ready for publication.

We highly appreciate the reviewer's positive opinion regarding our manuscript. We tried our best to polish it and hope it can provide enough contribution to the research community.